# The role of CXCR3/LRP1 cross-talk in the invasion of primary brain tumors

Kevin Boyé[1,2], Nadège Pujol[1,2], Isabel D Alves[3], Ya-Ping Chen[4], Thomas Daubon [1,2,5,6], Yi-Zong Lee[4], Stephane Dedieu[7], Marion Constantin[1,2], Lorenzo Bello[8], Marco Rossi[8], Rolf Bjerkvig[5,6], Shih-Che Sue[4], Andreas Bikfalvi[1,2] & Clotilde Billottet[1,2]

CXCR3 plays important roles in angiogenesis, inflammation, and cancer. However, the precise mechanism of regulation and activity in tumors is not well known. We focused on CXCR3-A conformation and on the mechanisms controlling its activity and trafficking and investigated the role of CXCR3/LRP1 cross talk in tumor cell invasion. Here we report that agonist stimulation induces an anisotropic response with conformational changes of CXCR3-A along its longitudinal axis. CXCR3-A is internalized via clathrin-coated vesicles and recycled by retrograde trafficking. We demonstrate that CXCR3-A interacts with LRP1. Silencing of LRP1 leads to an increase in the magnitude of ligand-induced conformational change with CXCR3-A focalized at the cell membrane, leading to a sustained receptor activity and an increase in tumor cell migration. This was validated in patient-derived glioma cells and patient samples. Our study defines LRP1 as a regulator of CXCR3, which may have important consequences for tumor biology.

[1] INSERM U1029, Pessac, 33615, France. [2] Université de Bordeaux, Pessac, 33615, France. [3] CBMN, UMR 5248 CNRS, Pessac, 33615, France. [4] Institute of Bioinformatics and Structural Biology, NTHU, Hsinchu, 30055, Taiwan. [5] K.G. Jebsen Brain Tumour Research Centre, Department of Biomedicine, University of Bergen, Bergen, 5009, Norway. [6] Department of Oncology, Luxembourg Institute of Health, Luxembourg, L-1526, Luxembourg. [7] CNRS UMR 7369 MEDyC, Université de Reims Champagne-Ardenne, Reims, 51687, France. [8] Neurosurgical Oncology Unit, Department of Oncology and Hemato-Oncology, Università degli Studi di Milano, Humanitas Resarch Hospital, Milan, 20089, Italy. Correspondence and requests for materials should be addressed to A.B. (email: andreas.bikfalvi@u-bordeaux.fr) or to C.B. (email: clotilde.billottet@u-bordeaux.fr)

The CXC chemokine receptor CXCR3 belongs to the family of G-protein-coupled receptors (GPCRs) that mediate diverse biological functions upon extracellular stimuli. CXCR3 has been reported to interact with various CXC chemokines (CXCL9-11, CXCL4, and CXCL4L1). Increase in CXCR3 expression has been found in many human tumors and has been correlated with poor prognosis in patients with breast cancer, colon cancer, glioma and osteosarcoma[1–4].

Three distinct CXCR3 spliced isoforms have been described including CXCR3-A, CXCR3-B, and CXCR3-alt, CXCR3-A and CXCR3-B being the most important. CXCR3-B displays a longer amino-terminal domain than CXCR3-A[5]. CXCR3-A is reported to promote cell proliferation, survival and migration, while CXCR3-B mediates growth inhibitory activity and apoptosis[6–9]. In renal carcinoma cells, the relative expression of CXCR3-A and CXCR3-B determines the effect on cell proliferation and survival and overexpression of CXCR3-B significantly inhibits cell proliferation and promotes apoptosis[6]. In gastric cancers, overexpresssion of CXCR3-B correlates with favorable prognosis[10]. Thus, CXCR3-A appears to mediate "switch on" signaling while CXCR3-B appears to mediate "switch off" signaling in tumors.

CXCR3 increases intracellular calcium levels and activates multiple signaling pathways, related to actin reorganization, proliferation, chemotactic migration, invasion, and cell survival. Many reports have shown that in various cell types (pericytes, endothelial cells, myofibroblast, T cells, epithelial cells, and tumor cells), binding of CXC chemokines to CXCR3 induces activation of p38 and ERK/mitogen activated protein kinases (MAPK), phosphatidylinositol 3-kinase (PI3K), and phospholipase C (PLC)[11–14].

Plasmon waveguide resonance (PWR) has been demonstrated ideal to follow GPCR activation and first signaling events[15, 16]. This method is highly sensitive and allows direct assessment of binding affinity and kinetics. Additionally, it can follow the orientation of anisotropic-oriented samples.

Lipoprotein receptor-related protein-1 (LRP1) is a large multiligand endocytic receptor that belongs to the low-density lipoprotein receptor family[17]. Members of this family were thought to be exclusively involved in receptor-mediated uptake of extracellular molecules, but many studies have revealed new roles of this family of receptors. LRP1 is widely expressed in several cell types including fibroblasts, neurons, astrocytes, macrophages, smooth muscle cells, and tumor cells. LRP1 is synthesized as a 600-kDa precursor protein that interacts with the ER chaperone receptor-associated protein (RAP)[18] and is processed into an extracellular ligand-binding subunit of 515 KDa (α chain) and a transmembrane (TM) and intracellular subunit of 85-kDa (β chain). The α-chain contains four extracellular ligand-binding domains (I–IV), and the intracellular domain of the β chain binds several adaptor proteins for efficient endocytic trafficking and signaling[19]. LRP1 is reported to regulate the abundance or the function of receptors/cell signaling proteins in the plasma membrane, including uPAR, EphA2, and neuropilin-1[20–23]. LRP1 has been reported to mediate endothelial and megakaryocyte cells responses to the CXC chemokine CXCL4[24, 25]. Mice lacking LRP1 in smooth muscle cells show greatly diminished vessel integrity[26]. Nakajima et al.[23] further demonstrated that LRP1 modulates the GPCR sphingosine-1-phosphate (S1P) signaling but does not interact with GPCR S1P. LRP1 is also critically involved in many processes that drive tumorigenesis and tumor progression[27].

In this article, we studied the status of CXCR3-A in tumor cells and new observations for receptor conformation, function, and regulation are presented. Furthermore, PWR provided new insights into CXCR3 activation at a structural level. One of our main finding is the involvement of LRP1 in the regulation of CXCR3 bioactivity and trafficking. This has significance for tumor infiltration at a fundamental and clinical level. This study reports an interaction between a classical CXC chemokine receptor and LRP1, and shows a regulatory role of LRP1 in GPCR conformation, trafficking, and biological activity in tumor cells.

## Results

**Expression and function of CXCR3 in glioma cells**. We investigated the expression of CXCR3 isoforms in different glioma cell lines (Supplementary Fig. 1A) by quantitative real-time PCR to distinguish CXCR3-A from CXCR3-B. Glioma cell lines express both CXCR3-A and CXCR3-B messenger RNA (mRNA) (Fig. 1a). Endogenous CXCR3 protein levels measured in all glioma cell lines by immunoblotting were variable (Fig. 1b). We used the U87 glioma cell line in the subsequent studies since it expresses low CXCR3 levels (Fig. 1b) and can, therefore, be engineered with adequate expression vectors. To reinforce our results, we used, in addition, the HEK-293 cell line which also expresses low CXCR3 levels (Fig. 1b).

We focused our study on CXCR3-A function and stably transfected U87 cells with vectors encoding CXCR3-A (U87-CXCR3-A). CXCR3-A expression was validated in transfected cell lines (Supplementary Fig. 1). U87-CXCR3-A cells migrated more (five-fold increase) in response to the CXCR3 agonist (PS372424) in comparison to empty-vector transfected control cells (U87-CTRL) (Fig. 1c). The increase in cell migration was inhibited by the treatment using a CXCR3 antagonist (SCH546738) (Fig. 1c). To reinforce the results, we used the T98G cell line that expresses endogenously significant levels of CXCR3-A compared to U87 (Fig. 1a). CXCR3 silencing in these cells inhibited migration (Supplementary Fig. 2A).

To study tumor growth and metastasis in vivo, the chick embryo's chorioallantoic membrane (CAM) model was implanted with U87-CTRL and U87-CXCR3-A cells (Fig. 1d). U87-CXCR3-A tumors expressed high amounts of CXCR3-A in vivo compared to U87-CTRL tumors (Fig. 1d, e). U87-CXCR3-A tumors displayed invasion with tumor strands invading the chicken host tissue, in contrast to U87-CTRL, which were circumscribed and highly angiogenic (Fig. 1d). Accordingly, uPA and metalloproteinases expression levels were increased in U87-CXCR3-A tumors (Fig. 1e). Blockade of CXCR3-A by SCH546738 leads to a strong inhibition of tumor invasion in the CAM model (Supplementary Fig. 2B).

**Conformational and dynamic properties of CXCR3-A**. We next conducted PWR experiments. This technique is able to determine binding affinity and kinetics. Additionally, it can probe changes in the molecular orientation of anisotropic lipid membranes and their embedded proteins by the use of two polarizations, perpendicular (p-pol) and parallel (s-pol) to the sensor surface. These correspond, respectively, to the long and short axis of lipids and proteins (Fig. 2a)[28, 29]. PWR is a well-established method to follow ligand-induced receptor conformational changes[15, 16, 30, 31]. Herein, we have detached membrane fragments from cells in culture (for details see Methods section). Membrane fragments of U87-CTRL or U87-CXCR3-A cells immobilized onto silica (glass slides or PWR sensor) pretreated with polylysine were analyzed by fluorescence microscopy and PWR. U87 cell membrane fragments were adsorbed on the slide and cell fragments rather than whole cells containing the CXCR3-A were indeed captured by this method (Supplementary Fig. 3A). PWR spectra performed before and after cell adhesion on the sensor showed a large increase in resonance minimum position (about 100 mdeg) that can be explained by mass increase due to the capture of cell fragments (Fig. 2b). The magnitude of the spectral shift correlated

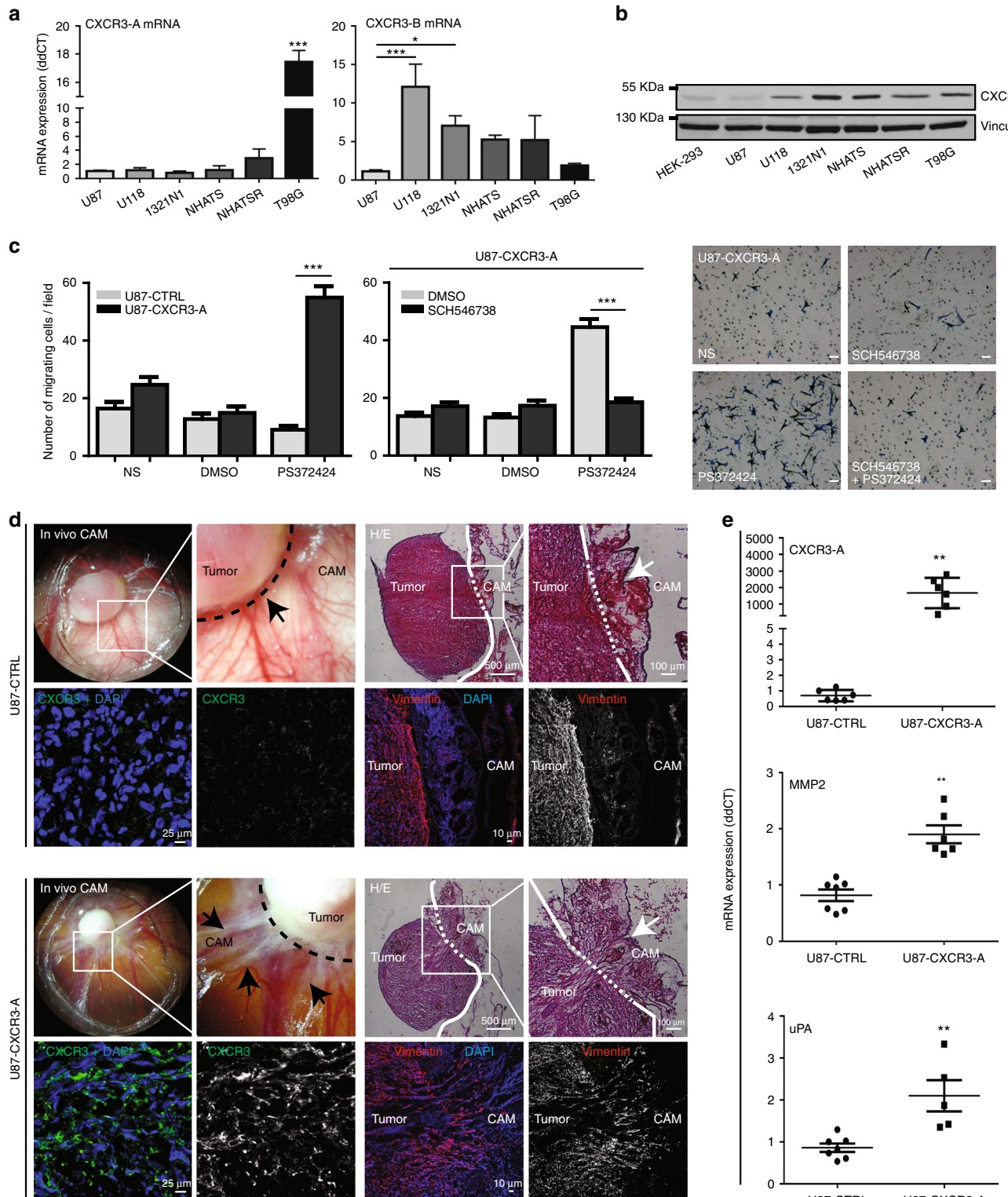

**Fig. 1** Expression and functional role of CXCR3-A in glioma. **a** Relative CXCR3 isoform mRNA expression in low (1321N1, NHATS, and NHATSR) and high-grade (U87, U118, and T98G) glioma. Values normalized to those obtained for U87 cell line. **b** CXCR3 protein expression in HEK-293 and glioma cell lines. Vinculin was used as a loading control. **c** Migration assays using CXCR3 agonist PS372424 (100 ng/ml) were performed in U87-CTRL and U87-CXCR3-A cells. U87-CXCR3-A cells were pretreated or not with CXCR3 antagonist SCH546738. NS non-stimulated condition, DMSO negative control. **d** CAM assays were performed using U87-CTRL (*n* = 21 eggs) or U87-CXCR3-A cells (*n* = 23 eggs). Seven days after implantation, images of CAM tumors with CXCR3 immunostaining and HE staining with corresponding vimentin immunostaining were performed in U87-CTRL and U87-CXCR3-A. **e** Relative CXCR3-A, MMP-2, or uPA mRNA expression in U87-CTRL and U87-CXCR3-A tumors (*n* = 7 eggs for each condition). All the results from three independent experiments were combined to calculate mean and SEM, and values were normalized to those obtained for the control cells. \*\*\**P* < 0.001, \*\**P* < 0.01, \**P* < 0.05. Multiple comparisons were performed with one-way analysis of variance, followed by Bonferroni post hoc tests. Statistical comparison between two groups was performed using the Mann–Whitney *U*-test

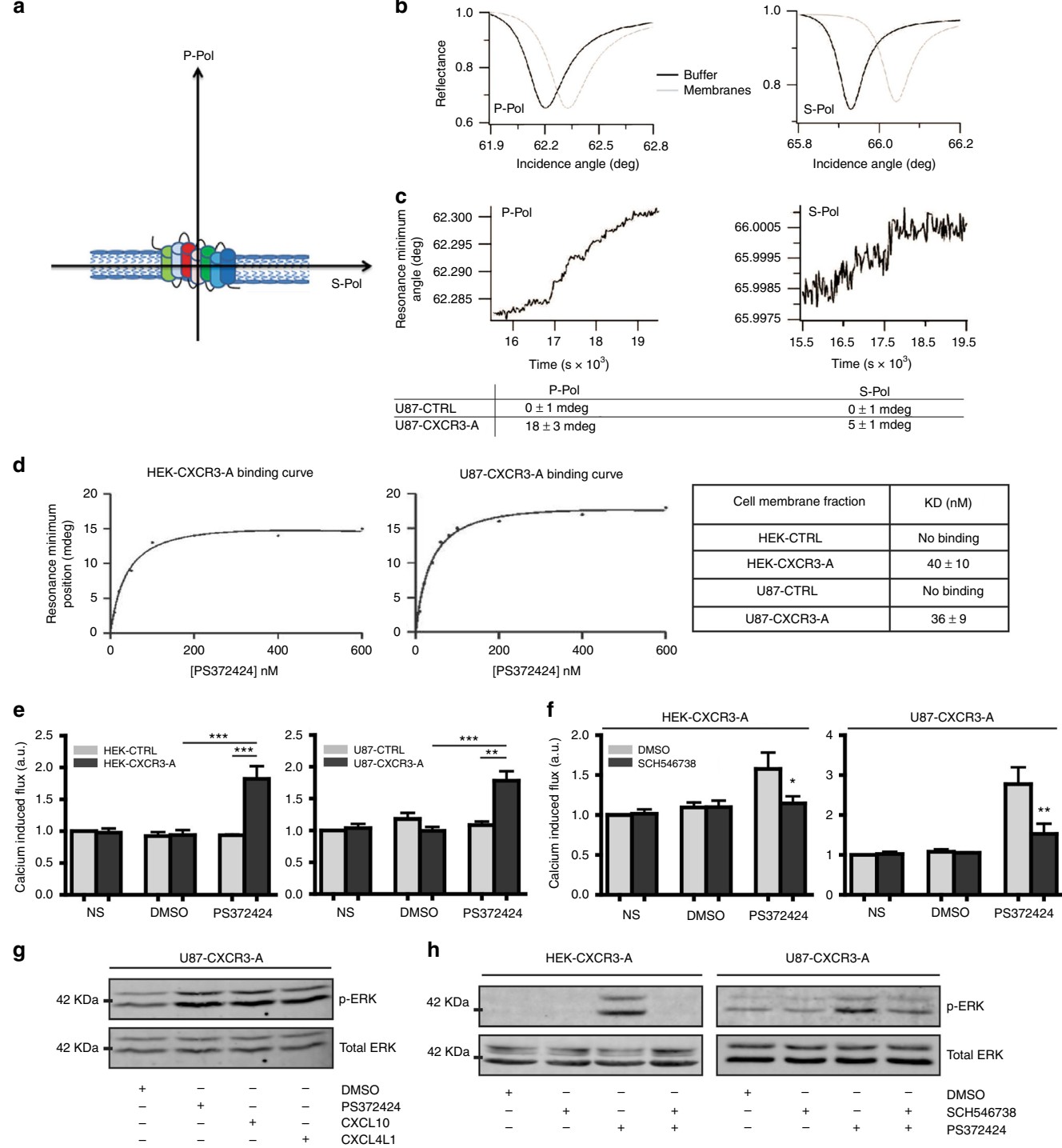

**Fig. 2** Conformational dynamic properties and activation of CXCR3-A. **a** Schematic representation of the *p*-polarized light perpendicular to CXCR3 (referred as *P*-Pol for elongation of the receptor in the plasma membrane) and of the *s*-polarized light parallel to CXCR3 (referred as *S*-Pol for spreading of the receptor along the plasma membrane). **b** PWR spectra for the PWR sensor before and after immobilization of the cell fragments obtained with *p*-(left panel) and *s*-(right panel) polarized light in U87-CXCR3-A cells. **c** Changes in the minimum resonance position following incremental addition of CXCR3 agonist PS372424 to U87-CXCR3-A cell fragments as a function of time obtained with *p*-(left panel) and *s*-(right panel) polarized lights. **d** Binding curve and $K_D$ values for PS372424 interaction with CXCR3-A in HEK-CXCR3-A and U87-CXCR3-A. **e** Relative calcium-induced flux in HEK-CTRL/HEK-CXCR3-A cells (left panel) and in U87-CTRL/U87-CXCR3-A cells (right panel) after 20 s of CXCR3 agonist PS372424 stimulation. **f** Relative calcium flux in non-stimulated (NS) and stimulated HEK-CXCR3-A cells (left panel) and U87-CXCR3-A cells (right panel) when cells were pretreated or not with CXCR3 antagonist SCH546738. DMSO negative control, a.u. arbitrary unit. ***$P<0.001$, **$P<0.01$, *$P<0.05$. **g** CXCR3 activation assessed by measuring the phosphorylation level of ERK1/2 after 5 min of stimulation (100 ng/ml of CXCR3 agonists: PS372424, CXCL10, or CXCL4L1). **h** Activation of MAPK/ERK pathways after CXCR3 stimulation when cells were pretreated or not with CXCR3 antagonist SCH546738. Total ERK was used as loading control. All the results represent three independent experiments combined to calculate mean and SEM. Values were normalized to those obtained for the control cells. Multiple comparisons were performed with one-way analysis of variance, followed by Bonferroni post hoc tests

well with that observed for a pure lipid membrane across the entire PWR sensor[29] and our results are comparable to the anisotropic response found for other GPCRs[31], although the deposition method was different. The addition of the agonist PS372424 to the U87-CXCR3-A cell fragments captured on the sensor led to spectral changes for both polarizations, but especially for the *p*-polarization (Fig. 2c) and no spectral changes were observed with U87-CTRL cell fragments. This reflects conformational changes of CXCR3-A along the receptor longitudinal axis (elongation) and indicate that ligand binding is specific.

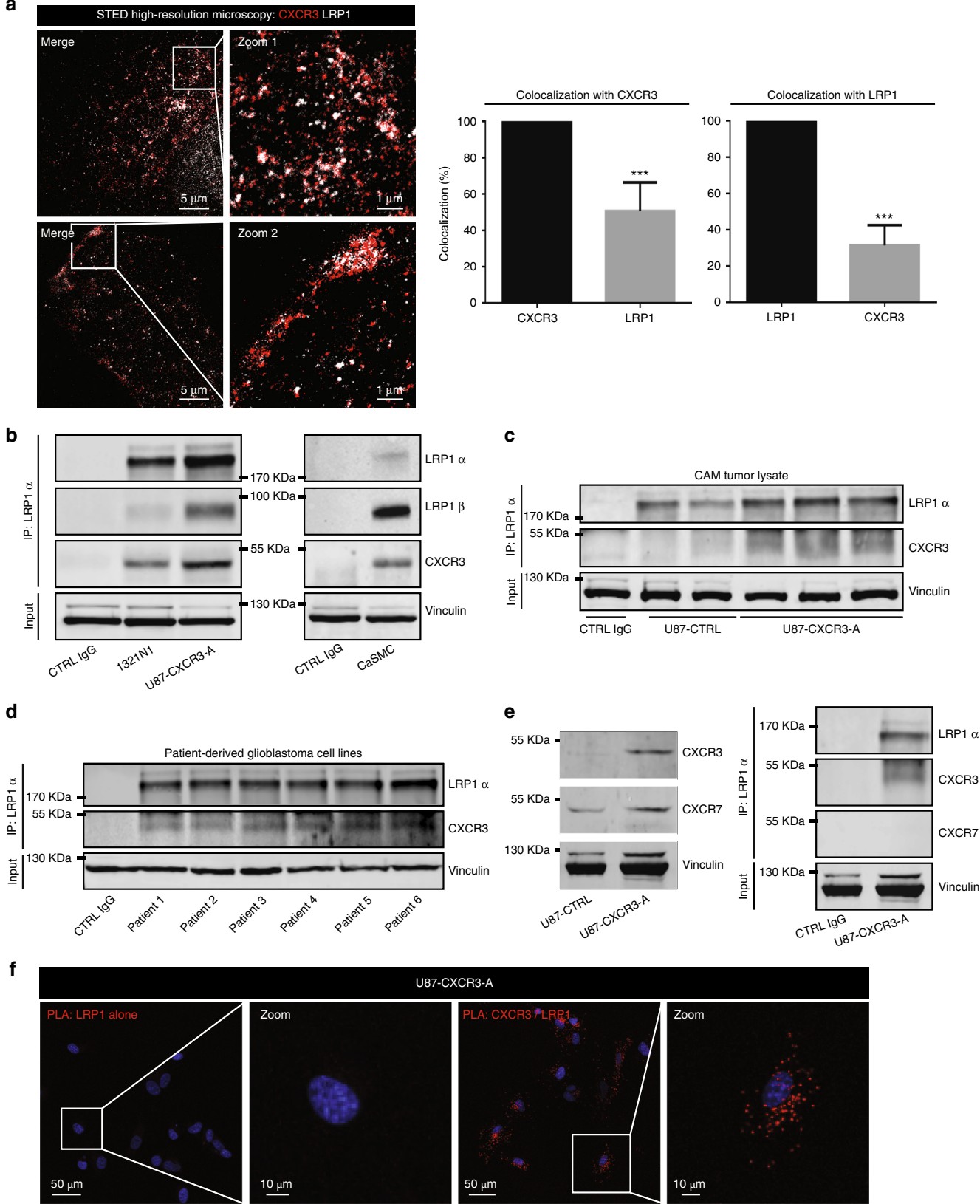

Moreover, binding of PS372424 to U87-CXCR3-A membranes was saturable ($K_D$ of $36 \pm 9$ nM) (Fig. 2d). Similar receptor conformational changes and affinity were obtained with HEK-293 cells transfected with vectors encoding for CXCR3-A (HEK-CXCR3-A) (Fig. 2d; Supplementary Figs. 1, 3).

PS372424 is a three amino-acid fragment of CXCL10. Similar results were obtained by using full-length CXCL10 albeit the binding affinity was higher (see Supplementary Fig. 6A later in the article).

**Downstream activation of CXCR3-A in glioma cells.** GPCRs are critically regulated by β-arrestins, which not only desensitize G-protein signaling but also initiate a G-protein-independent wave of signaling pathways. Our results show that after 15 min of stimulation, β-arrestin 1, and β-arrestin 2 are localized at the cell membrane (Supplementary Fig. 4A). Stimulation with PS372424 led to a peak in intracellular calcium levels after 20 s only in cells expressing CXCR3-A and is inhibited by SCH546738 (Fig. 2e, f). PS372424, CXCL10, and CXCL4L1 increased phospho-Erk1 and phospho-Erk2 in U87-CXCR3-A cells after 5 min of stimulation and this was inhibited by SCH546738 (Fig. 2g, h; Supplementary Fig. 3F, G).

**First evidence for an interaction between CXCR3 and LRP1.** We next investigated a possible link between CXCR3 and LRP1. We demonstrated by stimulated emission depletion (STED) microscopy that both CXCR3-A and LRP1 were present in the cytoplasm near the nucleus and at the membrane, and colocalized perfectly in the super resolution space (100 nm) of the STED (Fig. 3a, left panel; Supplementary Fig. 4B). We showed that in basal condition 55% of LRP1 colocalized with CXCR3-A and 30% of CXCR3 with LRP1 (Fig. 3a, right panels). This does not take into account juxta-positioned CXCR3 and LRP1 in which case the values would be higher. Double staining with antibodies against CXCR3 and LRP1 confirmed colocalization of LRP1 with CXCR3-A especially at the perinuclear area in basal condition (Pearson's correlation > 0.5) (Supplementary Fig. 4C). Upon stimulation with PS372424, both receptors were localized at the cell membrane, and Pearson's correlation was increased (Supplementary Fig. 4C). Furthermore, CXCR3 and LRP1 were co-immunoprecipitated in U87-CXCR3-A cells (Fig. 3b).

To validate this interaction in cell expressing endogenous receptors, we used the glioma cell line 1321N1 (Fig. 1b) and the coronary artery smooth muscle cells (CaSMC)[26]. In all cases, both LRP1 subunits and CXCR3 could be co-immunoprecipitated (Fig. 3b). The CXCR3 and LRP1 interaction was also evidenced in vivo in U87-CXCR3-A CAM tumors and in patient-derived glioblastoma cells (Fig. 3c, d). In comparison, CXCR7 known to be involved in glioma development[32], is expressed in U87 cells but was unable to interact with LRP1 (Fig. 3e).

To reinforce these results, we performed proximity ligation assay (PLA) experiments. U87-CXCR3-A cells were incubated with specific antibodies that recognize CXCR3 and LRP1. These experiments clearly show that CXCR3 and LRP1 colocalize (Fig. 3f). Appropriate controls were done which ascertained the specificity of the results.

**CXCR3-A binds LRP1 through its ligand-binding domain IV.** RAP known to antagonize ligand binding to LRP1 (domains I, II, and IV)[33] was able to significantly reduce CXCR3:LRP1 complex formation (Fig. 4a). This indicates that CXCR3-A interacts with the α chain of LRP1. HEK-CXCR3-A cells that only express low level of LRP1 were stably transfected with an HA-tagged mini-receptor derived from full-length LRP1 carrying only the ligand-binding domain IV (HA-LRPIV) (Fig. 4b). The CXCR3:LRP1 complex could also be immunoprecipitated in this case (Fig. 4b). These data demonstrate that LRP1 domain IV (LRPIV) is critically involved in CXCR3-A binding. To reinforce our data, we modeled the interaction between CXCR3- and the cystein-rich complement-type repeats (CR) of the LRPIV. The LRPIV contains CR 21-31. CR26 and CR27 have positive charged residues that may interact with the open ligand-binding pocket of GPCR, which contains many negatively charged residues[34, 35]. Separate molecular docking was performed between CXCR3-A and CR26 and CR27, respectively, using HADDOCK2.2[36, 37]. Only the HADDOCK output for the CXCR3-A:CR26 complex gave five clusters, while CR27 showed a less defined orientation for CXCR3-A binding. Positively charged residues within the CR26 N-terminus formed a continuously positive patch for the binding of CXCR3-A extracellular region. There are three basic residues (Arg, Lys, and Arg at position 3542, 3544, and 3547) found in the CR26 first and second β-sheets. A representative model is shown in Fig. 4c. The β1-loop-β2 was docked into CXCR3-A negatively charged pocket. The extracellular loop 2 (ECL2) in CXCR3-A has proper orientation for the interaction with the CR26 β1-loop-β2. Residues E196 and D195 in ECL2 are close to CR26 β1-loop-β2. This allows substantial attractive charge-based interactions.

Next, we modeled the interaction in the presence of the agonist PS372424 or full-length CXCL10 (Fig. 4d, e). Based on the model, CR26 engages CXCR3 mainly through the interaction with ECL2. In presence of PS372424, CR26 binding to CXCR3 is not overlapping with agonist binding to CXCR3, since PS372424 occupies only the inside of the GPCR hydrophobic pocket. Furthermore, CR26 is unable to penetrate deep into the binding pocket to interfere with agonist binding to CXCR3. Comparing the binding mode with that of the agonist, CXCL10 also displays a deep penetration into CXCR3 hydrophobic pocket. Furthermore, CXCL10 has a much wider binding surface and possesses more interactions with CXCR3, including an interaction with the receptor's N-terminus and the hydrophobic pocket. Therefore, we expect higher affinity of the full-length chemokine for CXCR3. This is indeed the case as our PWR results demonstrate ($K_D$ of $2.9 \pm 0.8$ nM for CXCL10 vs. $K_D$ of $36 \pm 9$ nM for PS372424; Supplementary Fig. 6 below).

**Fig. 3** Interaction of CXCR3 with LRP1. **a** Super resolution microscopy (STED) for U87-CXCR3-A cells (CXCR3 λ = 594 nm, LRP1 λ = 647 nm). Zoom 1 represents the perinuclear area in the cell and zoom 2 the membrane area of the cell. Scale: 5 and 1 μm (zoom). ×1000 magnification under a Leica SP8-STED microscope. The graph depicts the percentage of colocalization of each fluorophore with CXCR3 or LRP1 (n = 30). ***P<0.001. Statistical comparison between two groups was performed using the Mann–Whitney U-test. **b** Immunoprecipitation against LRP1 (IP LRP1α, antibody LRP1 8G1 for glioma cells, and antibody LRP1 D2642 for CaSMC) and western blot of the CXCR3:LRP1 complex (antibody 8G1 for LRP1α and 5A6 and 2703-1 for LRP1β) in glioma cells and coronary artery smooth muscle cells (CaSMC). CTRL IgG negative control of beads with IgG. **c, d** Immunoprecipitation against LRP1 (IP LRP1α, antibody LRP1 8G1) and western blot of the CXCR3:LRP1 complex (antibody 8G1 for LRP1α) in U87-CTRL and U87-CXCR3-A tumors implanted in the CAM (**c**) and in several GBM cell lines derived from patients (**d**). **e** Western blot probed with anti-CXCR7 and anti-CXCR3 antibodies in U87-CXCR3-A and immunoprecipitation against LRP1 (IP LRP1α, antibody LRP1 8G1). The experiments have been repeated three times with identical results. **f** Proximity ligation assay (PLA) in U87-CXCR3-A cells between CXCR3 and LRP1 using anti-CXCR3 (mouse 2 ar1) and anti-LRP1 antibodies (2703-1, rabbit monoclonal antibody for LRP1β)

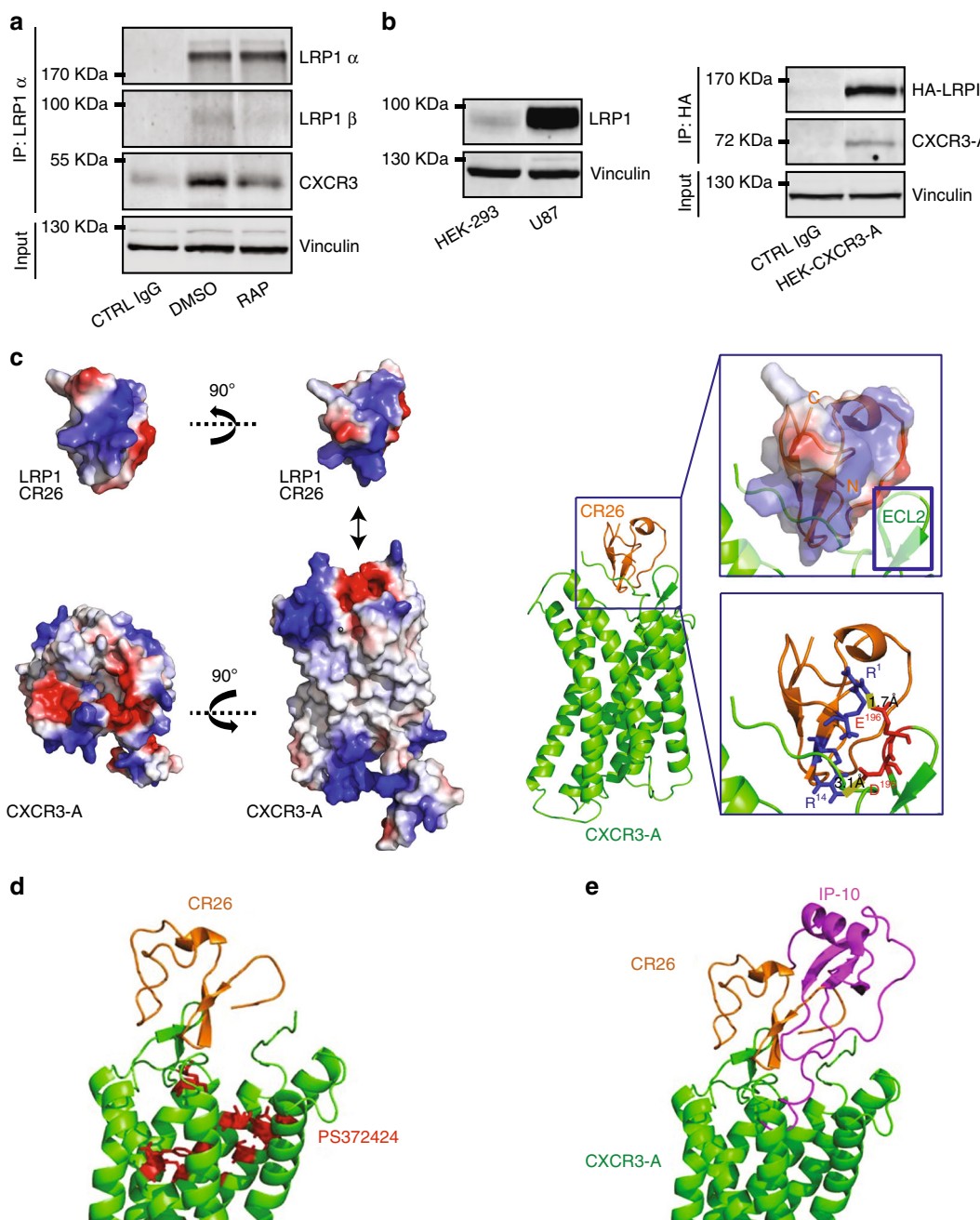

**Fig. 4** Interaction domain and modeling of CXCR3:LRP1 complex. **a** Immunoprecipitation against LRP1 (IP LRP1α, antibody 8G1) and western blot of the CXCR3:LRP1 complex (antibody 8G1 for LRP1α and 2703-1 for LRP1β) in U87-CXCR3-A cells pretreated or not with the LRP1 antagonist RAP. DMSO negative control. **b** LRP1 protein expression in HEK-293 and U87 cells (antibody 2703-1 for LRP1β) (left panel). Immunoprecipitation for the mini-LRP1 carrying domain IV (HA-LRPIV) in HEK-CXCR3-A cells transfected with the pCDNA.3 HA-LRPIV vector (right panel). The experiments have been repeated three times with identical results. **c** Binding orientation of CXCR3-A and LRP1-CR26 according to their electrostatic potentials. Red: negative charge and blue: positive charge (left panel). Docking model for ECL2 of CXCR3 and CR26 of LRP1 (right panel). **d** and **e** Structural interaction model of CXCR3-A and its ligand PS372424 (**d**) and CXCL10 (**e**)

**Clathrin-dependent internalization and recycling**. We next aimed at studying CXCR3 trafficking because LRP1 is an endocytic receptor. To investigate clathrin-dependent and independent pathways, we performed studies using antibodies for clathrin, for early endosomes (EEA1 and Rab5), for trans-Golgi recycling network (GOPC and syntaxin 6) or for lysosomal degradation pathway (Lamp-1). In unstimulated U87-CXCR3-A cells, colocalization of CXCR3-A with each marker was found near to the nucleus (Fig. 5a). In agonist-stimulated U87-CXCR3-

A cells, colocalization was also seen at the plasma membrane for all markers, and the Pearson's correlation was increased for clathrin and EEA1 after 15 min of stimulation (Fig. 5a; Supplementary Fig. 4D). At 60 min of stimulation, CXCR3-A, LRP1, GOPC, and syntaxin 6 were mainly detected in the cytoplasm (Fig. 5a). CXCR3-A colocalized with GOPC and syntaxin 6, suggesting its recycling through the trans-Golgi network. The same results were obtained in the HEK-CXCR3-A cells with clathrin, EEA1, and Rab5 immunostaining at 15 min of agonist

stimulation and with GOPC and syntaxin 6 immunostaining at 60 min of stimulation (Supplementary Fig. 5A). Immunogold labeling of ultrathin cryosections of U87-CXCR3-A cells demonstrated that CXCR3 and syntaxin 6 are both present in the trans-golgi network. These results confirm the localization of CXCR3 in the recycling compartment (Fig. 5b).

Furthermore, silencing of clathrin (Fig. 6a, b) demonstrated that CXCR3-A remained at the cell membrane under unstimulated conditions and after 60 min of stimulation (Fig. 6c–f). This was confirmed by inhibition of clathrin by Pitstop2, even if this inhibitor has some non-specific effects[38]. After 3 min of stimulation, a sustained increase in intracellular calcium was observed (Fig. 6g). We demonstrated that CXCR3-A is internalized with LRP1 in a clathrin-dependent manner to EEA1-positive early endosomes. No colocalization was found with caveolin and late endosomes or lysosomes (Supplementary Fig. 5B), which indicated that endocytosed CXCR3 receptors were protected from lysosomal degradation. This indicates that CXCR3 and LRP1 use a recycling pathway leading to a slower receptor turnover.

**LRP1 as a regulator of CXCR3 activity and function.** We studied whether LRP1 is involved in CXCR3-A conformational dynamics and trafficking. Silencing of LRP1 expression was performed in U87-CTRL and U87-CXCR3-A cells using three interference RNAs against human LRP1 with strong and similar efficiency (90% of LRP1 silencing) (Fig. 7a). In small interfering RNA (siRNA) LRP1-treated cells and stimulated with PS372424, both the magnitude (spectral shifts reported in mdeg) and the anisotropy of the response (ratio of the spectral shifts observed with the *p*- ad *s*-pol light) are altered (Fig. 7b; Supplementary Fig. 3D). The magnitude of the spectral changes induced by the ligand in siRNA LRP1 U87-CXCR3-A-treated cells was more than two-fold (45 and 25 mdeg for *p*- and *s*-polarization, respectively) when compared to cells treated with control siRNA (19 and 6 mdeg for *p*- and *s*-polarization, respectively) (Fig. 7b). This is correlated with a higher amount of receptors present in the cell fragments (Fig. 7c). Thus, the data support the idea that LRP1 depletion leads to receptor accumulation at the cell membrane and a ligand-induced conformational change. Also, the anisotropy of the ligand-induced receptor conformational change was greatly altered in siRNA LRP1-treated cells (Fig. 7b). This indicates that ligand-induced conformational change of the receptor is affected by the presence of LRP1 in the membrane. However, ligand affinity for the receptor is very much comparable to that observed in U87-CXCR3-A treated with control siRNA ($K_D$ of $38 \pm 9$ for siRNA LRP1 and $K_D$ of $41 \pm 11$ for siRNA control) (Supplementary Fig. 3E).

When using the full-length CXCL10 in these experiments, the affinity of the ligand was also not altered after LRP1 depletion ($2.9 \pm 0.8$ nM siRNA Control vs. $3.2 \pm 0.7$ nM siRNA LRP1) (Supplementary Fig. 6A). *P*-pol and *s*-pol values were strikingly modified and increased of about 60–50% for *s*-pol and *p*-pol, respectively, after LRP1 depletion (Supplementary Fig. 6A).

Flow cytometry and western blot for CXCR3 on subcellular membranous fractionation confirmed the increase of CXCR3-A at the cell membrane when LRP1 is depleted (Fig. 7d, f). Furthermore, modifications in the subcellular localization of CXCR3 were observed when LRP1 was depleted. LRP1 silencing showed that CXCR3-A accumulated at the cell membrane from the rear to the front and along membrane ruffles (Fig. 7e; Supplementary Fig. 6B). Clathrin was still recruited and colocalized with CXCR3-A at the plasma membrane in the absence of LRP1 (Fig. 7g). This indicates that internalization of CXCR3-A is strongly inhibited in the absence of LRP1 and

associated with impaired clathrin-coated vesicle formation (Fig. 7g).

In U87-CXCR3-A cells silenced for LRP1, we observed that the increase in intracellular calcium level was not only present 20 s after stimulation with the CXCR3 agonist but was sustained for a prolonged period of time (up to 3 min, Fig. 8a). The same experiments were performed in the glioma cell line T98G that expresses endogenously significant levels of CXCR3-A compared to U87 cells (Fig. 1a). CXCR3-A activation was observed 20 s after stimulation and sustained 1 min post stimulation when LRP1 was silenced (Fig. 8b). In addition, a marked phosphorylation of Erk1/2 was still observed in U87-CXCR3-A cells at 60 min of stimulation when LRP1 was depleted (Fig. 8c, left panel). The same results were obtained using RAP, known to competitively inhibit LRP1-dependent endocytosis (Fig. 8c, right panel). Finally, cell migration was increased in agonist-stimulated U87-CXCR3-A cells after LRP1 depletion (Fig. 8d, left panel) or LRP1 inhibition using RAP (Fig. 8d, right panel). Since the inhibition of LRP1 leads to cell invasion, we further studied the expression of LRP1 in U87-CTRL and U87-CXCR3-A CAM tumors. Co-immunostainings for vimentin (to label human U87 cells) and for LRP1 demonstrated that LRP1 expression was strongly downregulated in the invasive part of U87-CXCR3-A CAM tumors (Fig. 8e).

**LRP1 and CXCR3 expression in glioma samples from patients.** CXCR3 and LRP1 are expressed in low- and high-grade gliomas[39]. Data from global data base-derived analyses do not take into account the regional heterogeneity of gliomas. We therefore analyzed the regional distribution of CXCR3 and LRP1 by focusing on high-grade gliomas (glioblastomas, GBM). To this aim, biopsy spheroids of the human P3 GBM were implanted into the brain of immunodeficient mice to assess in vivo the expression of LRP1 and CXCR3. This experimental mouse intracranial GBM model recapitulates the regional heterogeneity of gliomas (Fig. 9a)[40]. Vimentin staining was performed to localize human P3 GBM cells in the mouse brain. We demonstrated that LRP1 is strongly expressed in the angiogenic part of the tumor, while it is decreased in the invasive part of the tumor (Fig. 9b, c). Regarding CXCR3 expression, reliable antibodies are lacking as they are produced in the mouse and cross-react with mouse tissue in the xenograft mouse model. Additionally, patients samples were taken from central and peripheral areas of the tumor based on intraoperative neuronavigated MR imaging sample collection (Fig. 9d). In the angiogenic core of the tumors, CXCR3 and LRP1 were both detected (Fig. 9e). LRP1 was localized in tumor, stroma, and vascular cells, whereas CXCR3 was only detected in tumor cells and localized in the cytoplasm (Fig. 9e). In the invasive part of the tumor, LRP1 expression was completely shut down while CXCR3 was still present in tumor cells. Most importantly, CXCR3 was localized at the cell membrane in this case (Fig. 9e). These results are in agreement with our mechanistic studies where we modulated the expression of LRP1 and CXCR3 in glioma cells. Taken together, our results are in favor of a role of CXCR3 and LRP1 in tumor cell invasion and infiltration, where LRP1 is downregulated at the invasive areas to allow accumulation of CXCR3 at the cell membrane leading to a sustained CXCR3 activation and increased tumor cell invasion.

## Discussion
Several studies have reported a role of CXCR3 in tumor growth and metastasis[1–3, 9]. However, the presence of three distinct spliced isoforms leads to functional diversity of CXCR3s.

In this article, we focused on the CXCR3-A isoform using glioma cells as a model system. Gliomas are classified according

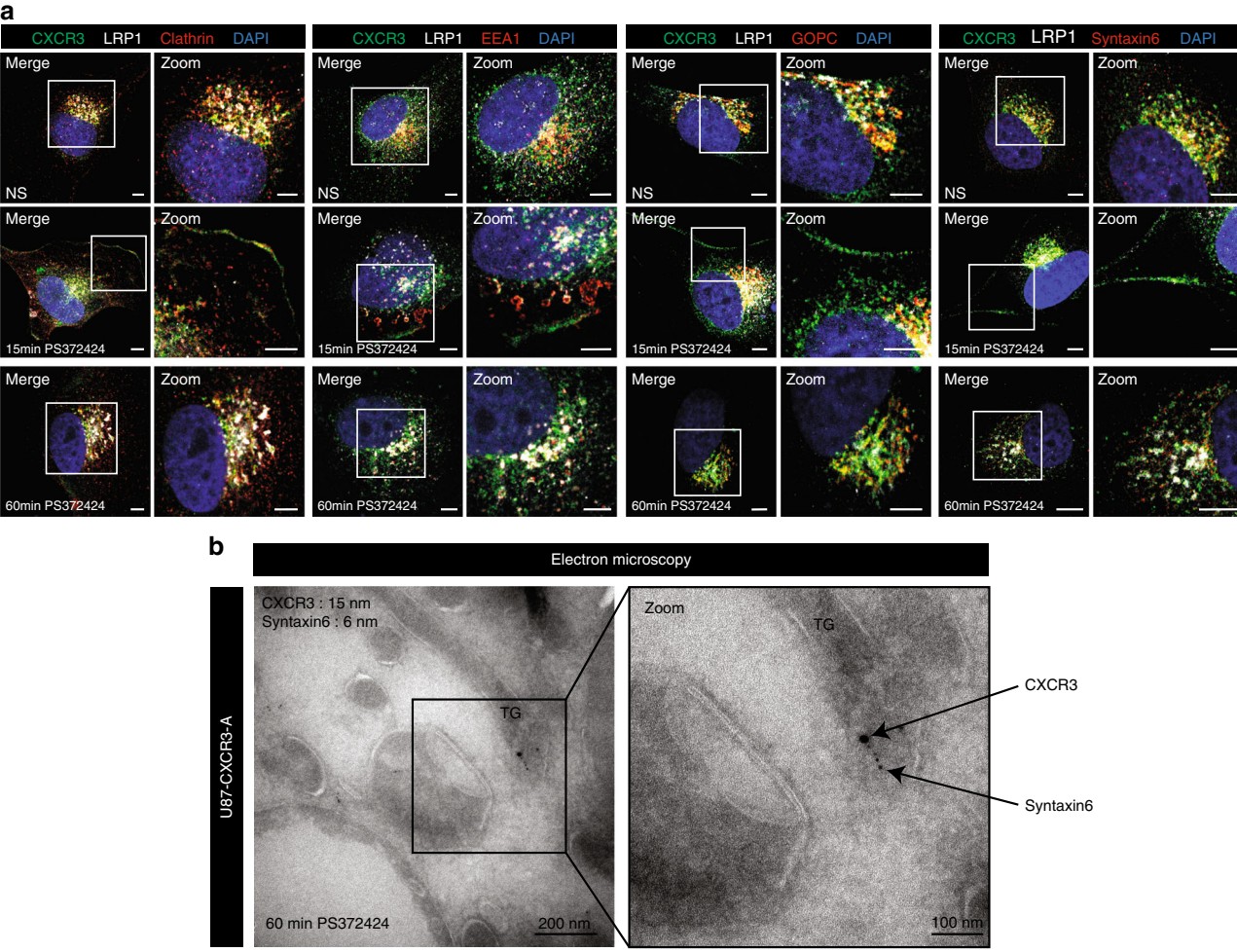

**Fig. 5** Internalization, trafficking and recycling of CXCR3-A. **a** Membrane or perinuclear localization of CXCR3 ($\lambda = 488$ nm) and colocalization with LRP1 ($\lambda = 647$ nm) in non-stimulated (NS) U87-CXCR3-A cells and upon stimulation with PS372424 at 15 and 60 min. Internalization and recycling of CXCR3 studied by immunostaining for clathrin, EEA1, GOPC, or Syntaxin6 (for each marker $\lambda = 547$ nm). Nuclei were stained with DAPI ($\lambda = 405$ nm). Scale: 5 μm. The experiments have been repeated three times with identical results. **b** Electron microscopy with immunogold labeling of CXCR3 (15 nm diameter) and Syntaxin6 (6 nm diameter) in ultrathin cryosections of U87-CXCR3-A cells stimulated with PS372424 for 60 min. TG trans-Golgi ($n = 3$ grids)

to World Health Organization (WHO)[41] from low- to high-grade gliomas. Grade IV tumors (glioblastoma multiforme, GBM) are highly malignant and usually resistant to chemotherapy. Glioblastomas are highly vascularized, proliferative, infiltrative, and associated with a poor prognosis[42]. Recent reports suggest an involvement of CXCR3 in glioma development[4, 43]. A strong correlation of CXCL10/CXCR3 expression with tumor grade in cultured cells derived from grade III astrocytomas and from grade IV glioblastomas has been reported[44]. However, neither the specific roles of the CXCR3 isoforms nor the mechanisms of activation and the regional variations of expression within the tumor have been investigated in these studies.

We found that CXCR3 isoforms mRNAs were expressed at variable levels in glioma cell lines of various grade as previously published[43]. It was also reported that high expression levels of CXCR3 conferred poor prognosis and could represent an independent prognostic marker for glioblastomas[4]. This is in contrast to TGCA and Sun et al.[39] data, where global expression CXCR3 does not impact on survival in GBM.

The functional role of CXCR3-A was studied in U87 glioma cells overexpressing CXCR3-A, in glioma cells that endogenously express high levels of CXCR3 and in cells from glioblastoma

patients. In vitro cell migration experiments and analysis in the chick embryo glioma model demonstrated that CXCR3-A increases cell glioma migration and invasion. CXCR3-A overexpressing tumors exhibited reduced angiogenesis but increased invasiveness with tumor strands migrating away from the main tumor mass in vicinity of CAM blood vessels. These results clearly show that CXCR3-A promotes glioma cell migration, which is consistent with a pro-invasive effect and is in agreement with the clinical data.

We used PWR to study the conformational dynamics and activity of CXCR3-A. The PWR results demonstrated that ligand-induced spectral changes lead to mass increases. In as much as the ligand by itself cannot explain such larger spectral changes, those correspond to receptor conformational changes with an overall increase in membrane mass density. Additionally, ligand-induced spectral changes were anisotropic. This reflects anisotropic conformational changes in the receptor with larger changes along the receptor longitudinal axis explained by receptor elongation. It is noteworthy that the conformational change observed here did not modify the affinity for the ligand. This implies that receptor elongation is not significantly affecting ligand affinity. Plasmon resonance studies on rhodopsin, which, like CXCR3

belongs to class A of the GPCR family, have demonstrated that upon light activation, the surrounding lipid environment suffers a considerable increase in membrane thickness[45]. This was confirmed by high-resolution structural studies where protein elongation was attributed to structural changes along the TM helix 5[46, 47].

Stimulation of CXCR3-positive glioma cells with the agonist led to a marked increase in calcium flux and in phospho-Erk1 and phospho-Erk2. These results are in agreement with earlier studies

that have shown that CXCR3 is able to elicit an increase in intracellular calcium level and activation of multiple signaling pathways[11–14, 48]. This correlates with the ability of CXCR3 to induce invasion and cell survival, compatible with a role in metastasis[1, 6].

Receptor internalization limits receptor availability at the cell surface, and thus ligand binding. When receptors reach endosomes, they can be sorted directed into the recycling pathway or targeting into the degradative pathway. Diverse mechanisms of

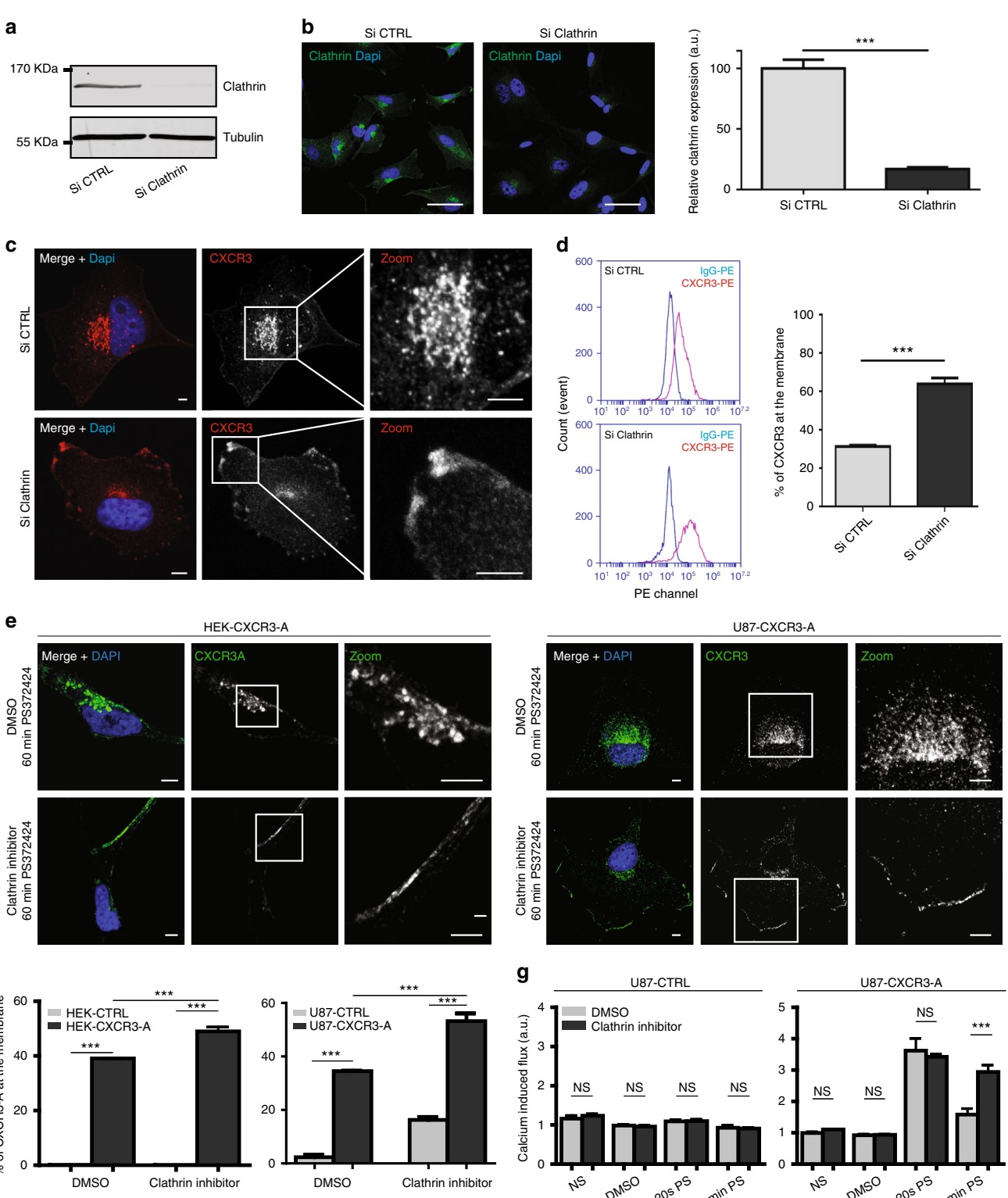

endocytosis have been reported for GPCR[49]. For CXCR3, most of the studies have been performed in T lymphocytes. The chemokine system is rather complex in these cells, as ligands of CXCR3 can preferentially activate distinct internalization pathways[50]. Meiser et al.[51] have described in detail endocytosis of CXCR3 in T lymphocytes after CXCL11 stimulation. They showed that CXCR3 is rapidly downregulated and that CXCR3 can be internalized independently of clathrin, β-arrestin 1/2. CXCR3 is directed into the degradative pathway, mediated in part by lysosomes and proteaosome, leading to long-term attenuation of signaling. Cell surface replenishment of CXCR3 occurred over several hours and is dependent upon de novo CXCR3 synthesis. This is in stark contrast with our study. We demonstrated that β-arrestin 1/2 was localized to the membrane just 15 min after agonist stimulation. CXCR3-A was rapidly internalized via clathrin-coated vesicles enriched with LRP1, and then recycled to the cell surface using retrograde trafficking from endosomes to the trans-Golgi network. A similar recycling mechanism has already been described for GPCR β1-adrenergic and GLUT4 receptors[52, 53]. We can only speculate about the reasons for these differences. Our study investigates CXCR3 kinetics in tumor cells and not immune cells. Furthermore, the difference may reside in the ligands used in both studies. Our contention is supported by Luker et al.[54] who showed that, in breast cancer cells, GPCR CXCR7 trafficking is regulated by β-arrestin 2 and that internalization is clathrin-dependent. They further demonstrated that CXCR7 was recycled to the cell membrane. CXCR7 has properties similar to those of CXCR3 in that it is known to promote tumor growth and metastasis in mammary carcinoma and glioma[32]. Goodman et al.[55] also showed that β-arrestin functions as an adaptor in the receptor-mediated endocytosis pathway, and suggest a general mechanism for regulating the trafficking of GPCRs. Plasma membrane localization and beginning of internalization of CXCR3-A were observed after short-term stimulation. The mechanisms of relocalization of CXCR3-A to the membrane after agonist stimulation is at present not known. Complete internalization with cytoplasmic localization of CXCR3-A was observed after long-term stimulation.

One of the most important aspects of our study is related to the involvement of LRP1 in CXCR3 conformation, activity and trafficking. LRP1 has been reported to be involved in tumor progression, in part through the modulation of the extracellular tumor environment[24, 27, 56]. LRP1 is essential for the integrity of the vascular wall through its capacity to regulate PDGFRβ-dependent activation of PI3K[26, 57]. It is, furthermore, critically involved in invasion and neovascularization, processes which drive tumor progression and metastasis[27]. LRP1 expression was detected in glioblastoma and to a lesser extent in lower-grade astrocytomas[58, 59]. In addition, LRP1 has been shown to mediate endothelial and megakaryocyte cells responses to the CXC chemokine CXCL4[24, 25] and to modulate the GPCR S1P signaling without interacting with S1P receptors but by restraining the activation of Gαi subunit[23]. In our study, LRP1 was critically

involved in CXCR3 activation and trafficking which is an entirely new finding for a classical CXC chemokine receptor. We provide furthermore strong evidence for a physical interaction between CXCR3 and LRP1, which may have a broad significance. This is based on the following: (1) high-resolution imaging demonstrate a close spatial proximity between CXCR3 and LRP1, (2) the complex is co-immunoprecipitated in CXCR3/LRP1 expressing cells, (3) the complex is present not only in tumor cells, but also in non-transformed cells such as smooth muscle cells, which have an important role in vessel maintenance and integrity, (4) the interaction between CXCR3 and LRP1 was inhibited by RAP, and (5) a model could be constructed for the CXCR3-A:LRP1 interaction involving in ECL2 of CXCR3 and CR26 of LRPIV.

Since, the interaction between CXCR3 and LRP1 was inhibited using RAP, CXCR3 should mainly interacts with the α chain of LRP1, known to bind other ligands as well such as uPA, CD44, and TIMP-1[60, 61]. We clearly demonstrated that the LRPIV is sufficient for the interaction between CXCR3-A and LRP1, which is also supported by the molecular docking analysis.

We also depleted or functionally silenced LRP1 using RAP. In this case, a considerable increase in the magnitude of the ligand-induced conformational change of CXCR3 and in receptor number at the membrane was observed. Moreover, the ratio of the spectral shifts obtained with $p$- and $s$-polarization (correlated with receptor anisotropy) was modified when LRP1 was depleted.

LRP1 is known to be largely confined to clathrin-coated pits, which suggests a role in clathrin-dependent internalization[60, 62]. We show that silencing of LRP1 inhibits CXCR3 internalization with an increase of CXCR3 at the cell membrane and that it is linked to an impairment in the formation of clathrin-coated vesicles. Furthermore, depletion of clathrin leads to the same effects, which shows that CXCR3 internalization is clathrin-dependent and downstream of LRP1.

This resulted in a sustained increase in calcium flux after agonist stimulation and Erk1/Erk2 phosphorylation after 60 min of stimulation. In addition, LRP1 depletion or functional silencing further increased cell migration of glioma cells.

CXCR3 and LRP1 are expressed in low- and high-grade gliomas[39]. These global data base-derived analyses do not take into account the regional heterogeneity of gliomas. Glioblastomas are heterogeneous tumors and are characterized by pronounced angiogenesis and tumor cell infiltration. Global analysis of the tumor transcriptome cannot evidence regional changes in expression that reflect tumor heterogeneity. We clearly show regional heterogeneity in glioblastoma when various areas from the angiogenic core and the invasive zone are analyzed. While CXCR3 is expressed in both, angiogenic and invasive areas, LRP1 expression is strongly reduced in invasive zones of the tumor. Most importantly, CXCR3 is localized at the tumor cell membrane in invasive areas, while in angiogenic parts of the tumor CXCR3 is mostly found at an intracellular location. This is in full agreement with our in vitro mechanistical studies. The inhibition of LRP1 expression in invasive tumor areas is an important

**Fig. 6** Clathrin-dependant internalization of CXCR3-A. Western blot (**a**) and immunostaining (**b**) for clathrin in U87-CXCR3-A cells treated with control siRNA (Si CTRL) or siRNA against human Clathrin heavy chain (Si Clathrin, smart pool of four siRNA, Dharmacon). Tubulin used as a loading control in the western blot. Scale: 50 μm. a.u. arbitrary unit. ***$P<0.001$. **c** Membrane or perinuclear localization of CXCR3 ($\lambda = 547$ nm) with Si CTRL or Si Clathrin. Scale: 5 μm. **d** FACS analysis of the percentage of CXCR3 at the membrane on U87-CXCR3-A cells treated with Si CTRL or Si Clathrin. ***$P<0.001$. **e** Clathrin-dependent internalization of CXCR3-A. Cells were preincubated with the clathrin inhibitor Pitstop 2tm. Analysis by immunofluorescence at ×630 magnification under confocal laser scanning microscope (Nikon eclipse Ti). Scale: 5 μm. **f** Constitutive internalization of CXCR3-A shown by FACS analysis in HEK-CTRL/HEK-CXCR3-A (left panel) and in U87-CTRL/U87-CXCR3-A (right panel) cells. **g** Cells were preincubated with the Pitstop 2tm and relative calcium flux assessed in U87-CTRL and in U87-CXCR3-A cells. NS non-stimulated condition, DMSO negative control, 20 s PS 20 s post stimulation, 3 min PS 3 min post stimulation, a.u. arbitrary unit, NS non-significant, ***$P<0.01$. All the results represent three independent experiments combined to calculate mean and SEM. Statistical comparison between two groups was performed using the Mann–Whitney U-test. Multiple comparisons were performed with one-way analysis of variance, followed by Bonferroni post hoc tests

finding. It may be of more general significance for tumor biology and may apply to other receptor systems with which LRP1 cooperates and other tumor types as well. These issues as well as the precise mechanism of silencing will be investigated in follow-up studies from our laboratory.

Taken together, we provide a number of solid arguments for a regulatory role of LRP1 in CXCR3 function. Our findings may also be of importance for glioma pathobiology (Fig. 10). Indeed, glioblastoma has a highly invasive behavior and high recurrence rate because of cells infiltrating the adjacent tissues during the evolution of the disease[41]. Thus, our finding may not only enrich

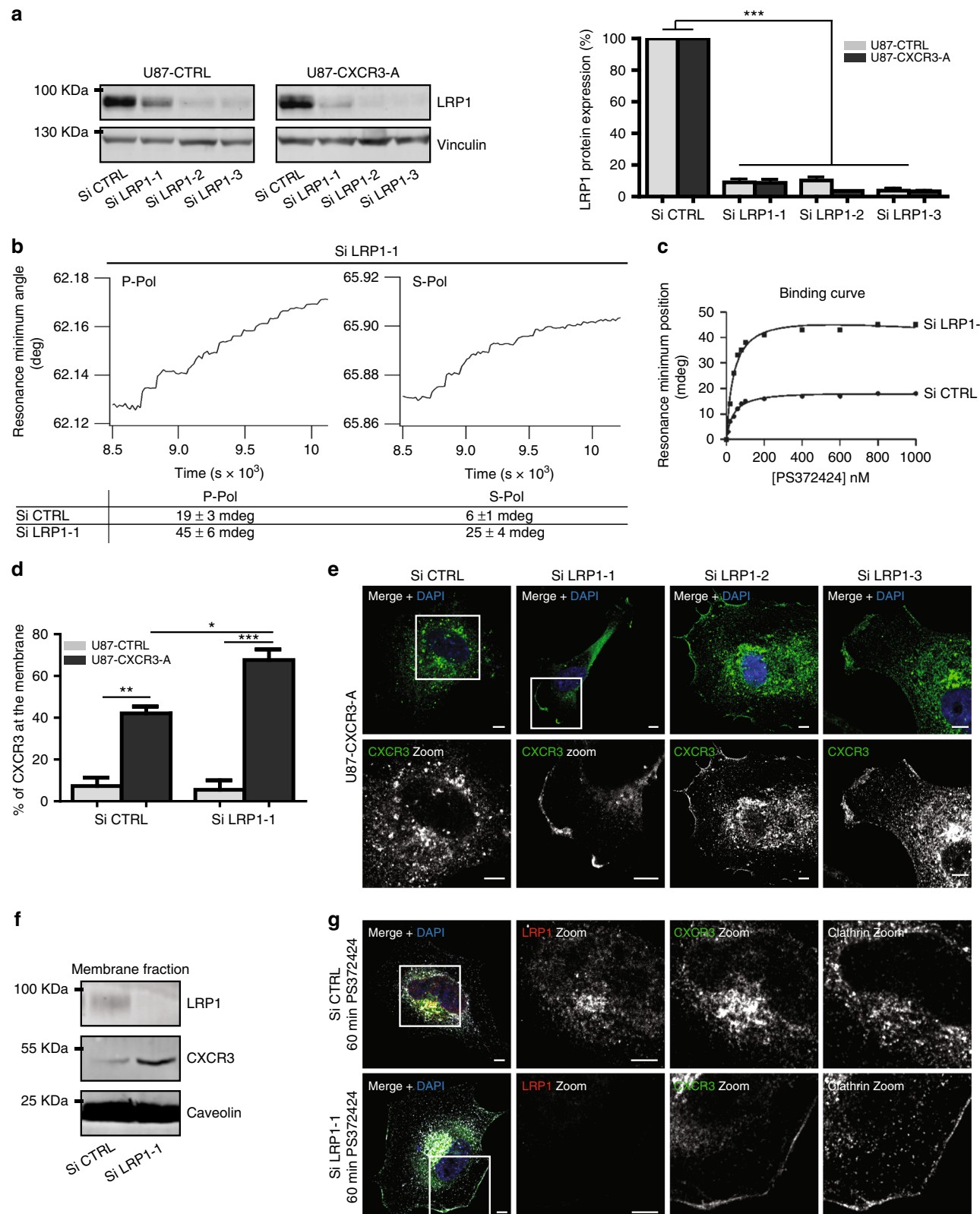

our knowledge of chemokine biology but also suggest strategy for therapeutic intervention.

## Methods

**Reagents and antibodies**. CXCR3 agonist (PS372424, ref: 239823) and recombinant human CXCL10 (ref: 300-12) were purchased from Calbiochem and Peprotech, respectively. CXCR3 antagonist (SCH546738) was synthesized and purchased from ChemExpress (Shanghai, CN). RAP generated as previously described[60]. Clathrin inhibitor Pitstop2, mouse monoclonal antibody against CXCR3 [2ar1] (ref: ab64714), mouse monoclonal antibody against CXCR3 [1C6] (ref: ab125255), rabbit polyclonal antibody against CXCR3 (ref: ab77907), rabbit monoclonal antibody against CXCR3 (ref: ab154845), rabbit monoclonal antibody against CXCR4 (ref: ab124824), rabbit polyclonal antibody against CXCR7 (ref: ab72100), mouse monoclonal antibody against LRP1 (5a6, epitope in the beta chain of LRP1) (ref: ab28320), and mouse monoclonal antibody against LRP1 (8G1, epitope in the alpha chain of LRP1) (ref: ab20384) were purchased from Abcam. Rabbit monoclonal antibody against LRP1 (ref: 2703-1, epitope in the beta chain of LRP1) was purchased from Epitomics and goat polyclonal antibody against LRP1 (ref: D2642, epitope in the alpha chain of LRP1) was a kind gift of Pr. Dudley Strickland (University of Maryland, Baltimore, USA). Mouse monoclonal antibody against GFP [B-2] (ref: sc-9996) was purchased from Santa Cruz Biotechnology. Purified Mouse Anti-β-Arrestin Clone 10/β-Arrestin1 (RUO) was purchased from BD Bioscience (ref: 610550). Vesicle Trafficking Antibody Sampler Kit (ref: 9765), Endosomal Marker Antibody Sampler Kit (ref: 12666), PathScan Multiplex western Cocktail I (ref: 5301S), and rabbit primary antibody against total p42/p44 MAPK (ref: 9102) were purchased from Cell Signaling Technology. Mouse monoclonal antibody against vinculin (ref: V9131) was purchased from Sigma-Aldrich. Mouse monoclonal antibody against human vimentin (ref: sc-6260 clone V9) was purchased from Santa Cruz.

**Migration assays**. Migration assays were performed on transparent polyethylene terephthalate (PET) membrane 8.0 μm pore size (ref: 353097, Dutscher). About 25,000 cells were seeded on the top part of the membrane and CXCR3 agonist (100 ng/ml) was added below in the well in serum-free medium. For migration experiments using RAP, cells were stimulated 24 h with RAP (5 μM) and then seeded on the top part of the membrane with CXCR3 agonist (100 ng/ml) and RAP (5 μM). For migration experiments using siRNA for LRP1, CXCR3 agonist gradient was performed with 100 ng/ml on top part of the membrane and 500 ng/ml below in the well. After 6 h, cells were fixed 10 min with 30% acetic acid and 40% methanol and then stained 15 min with 40% methanol, 30% acetic acid, and 0.5% coomassie blue. Cells which migrated through the membrane were then counted with an inverted epifluorescence microscope (Nikon Eclipse TE200, Obj ×10).

**Gene expression analysis**. Total RNA was isolated from cells using Trizol reagent (ref: TR118, Sigma-Aldrich) according to the manufacter's instruction. About 1 μg of total RNA was reverse-transcribed into complimentary DNA (cDNA) using the high-capacity cDNA reverse transcription kit (AbCam 4368814). The resulting cDNAs were amplified using specific primers for the genes of interest. GAPDH (glyceraldehyde-3-phosphate dehydrogenase) and HPRT (hypoxanthine guanine phosphoribosyl transferase, Fisher Scientific, Hs99999909_m1) were used as internal controls. We applied the real-time PCR method using the FG, POWER SYBR GREEN PCR (Fisher Scientific, 10658255) for CXCR3-A: forward sequence CCATGGTCCTTGAGGTGAGTG and reverse sequence AGCTGAAGTTCTCC-AGGAGGG, for MMP-2: forward sequence ccgtccccatcatcaa and reverse sequence aggtattgcactgccaactttt and for uPA: forward sequence cttaactccaa-cacgcaagggg and reverse sequence agcttgtgccaaactggggatc. Taqman univ PCR master mix (Fisher 10157154) was used for CXCR3-B: probe sequence (6FAM) CCCGTTCCCGCCCTCACAGG, forward sequence TGCCAGGCCTTTACA-CAGC and reverse sequence TCGGCGTCATTTAGCACTTG. Real-time PCR reactions were performed on a StepOnePlus Real-Time PCR system using StepOne software V2.3 (Applied Biosystems).

**Recombinant CXCL4L1 expression and purification**. CXCL4L1 was expressed in *Escherichia coli* BL21 (DE3) harboring an expression plasmid, pET-43.1a-CXCL4L1. Cells were cultured in LB media with 100 μg/ml ampicillin at 37 °C and subsequently induced by 1 mM isopropyl-thio-β-galactopyranoside (IPTG) for 4 h. Cells were harvested by centrifugation, stored at −20 °C and resuspended in lysis buffer (50 mM Tris-HCl, pH 7.4, and 100 mM NaCl), followed by sonication. After centrifugation, the pellet of recombinant CXCL4L1 was resuspended in lysis buffer containing 0.8% (v/v) Triton X-100 and 0.8% sodium deoxycholate with stirring for 20 min at room temperature, followed by centrifugation. This process was repeated three times. The final pellet was dissolved in denaturation buffer (6 M guanidinium hydrochloride, 50 mM Tris-HCl, pH 8.0, 500 mM NaCl, and 5 mM β-mercaptoethanol) and stirred 2 h at room temperature. The denatured CXCL4L1 was dialyzed with refolding buffer (0.9 M guanidinium chloride, 50 mM Tris-HCl, pH 8.0, 500 mM NaCl, 5 mM freshly added cysteine and methionine) at 4 °C for 6 h. The refolding buffer was replaced by the binding buffer (50 mM Tris-HCl, pH 7.4, and 500 mM NaCl). Precipitate of protein solution was removed by centrifugation. The protein solution containing folded CXCL4L1 was subjected to a heparin high-performance affinity column (ÄKTA FPLC system, GE Healthcare) equilibrated with binding buffer. Bound CXCL4L1 was eluted in 1 M NaCl and injected into a Superdex 75 HR 10/300 column (ÄKTA FPLC system) in 50 mM sodium phosphate buffer, pH 7.4, 150 mM NaCl. The eluted CXCL4L1 was further purified by HPLC system (Beckman Coulter Inc.) using a C18 reverse phase column (Phenomenex Inc.). The purified CXCL4L1 was lyophilized and kept frozen at −80 °C.

**Construction of CXCR3 expression plasmids**. The coding regions of human CXCR3-A cDNA were cloned in two consecutive steps. Human CXCR3-A cDNA fragments were amplified by PCR (forward primer: aaaacggtaccatggtccttg and reverse primer: aaaaggatcccaagcccgagt) from pEGFPN2-CXCR3-A. The CXCR3-A amplicons of 1125 bp were subcloned into the pCDNA3.1+ vector (Amersham Biosciences). Purified PCR products were digested with *Bam*HI and *Kpn*I restriction enzymes and inserted into the pCDNA3.1+ vector. Sequences of the plasmids were confirmed by DNA sequencing.

**Flow cytometry analysis**. Cells were washed three times with phosphate-buffered saline (PBS), 5% BSA and labeled in suspension with R-phycoerythrin (PE) mouse anti-human CD183 (ref: 557185BD, BD Pharmingen) or PE Mouse IgG1, Isotype Control (ref: 555749, BD Pharmingen) for 20 min at 4 °C. Cells were then washed three times with PBS, 5% BSA and analyzed on an AccuriC6 flow cytometer with BD AccuriC6 software (BD, San Jose, CA) as recommended to the manufacturer's instruction. A minimum of 10,000 events was recorded per sample.

**Small interfering RNA knockdown experiments**. For inhibition of CXCR3, Clathrin, and LRP1 expression, cells were transiently transfected with annealed siRNA. CXCR3, Clathrin, and LRP1-specific siRNAs were used (ON-TARGETplus Human CXCR3 (ref: 2833), ON-TARGETplus Human Clathrin heavy chain (smart pool of 4 siRNA, L-004001-01-0005), and ON-TARGETplus Human LRP1 (ref: 4035) with siRNA number 7 named siLRP1-3, GE Healthcare Europe GmbH, Dharmacon). A siRNA-random (siRNA CTRL, SR-CL000-005) used as a negative control. Two others siRNAs against human LRP1 were synthesized by Eurogentec (Angers, France). The LRP1-specific sense and antisense RNA oligonucleotides were as follows: siLRP1-1/sense siRNA, GCAGUUUGCCUGCAGAGAUtt, and antisense siRNA, AUCUCUGCAGGCA AACUGCtt and siLRP1-2/sense SiRNA AUGCUGACCCCGCCGUUGCtt and antisense siRNA GCAACGGCGGGGUCAGCAUtt. Transfection was performed using lipofectamine RNAimax Kit (Fisher 10601435) in accordance with the manufacturer's protocol, with siRNA at a final concentration of 20 nM in media without antibiotics. After transfection, cells were washed with PBS and fresh complete media was added for 48 h.

**Tumor spheroids and cell culture**. P3 GBM cells were taken and cultured from a patient biopsy[63]. Patient-derived cell lines were cultured as spheroids in neurobasal

**Fig. 7** LRP1-dependent conformational dynamics and internalization of CXCR3-A. **a** LRP1 downregulation in U87-CTRL and U87-CXCR3-A cells. Western blot for LRP1 in cells treated with three different siRNA against human LRP1. Vinculin as a loading control. **b** Changes in the minimum resonance position following incremental addition of PS372424 to the cell fragments of U87-CXCR3-A cells treated with LRP1 SiRNA (Si LRP1-1) or control SiRNA (Si CTRL) as a function of time (p-polarization and s-polarization light). **c** Binding curve for PS372424 interaction with the CXCR3-A following Si CTRL (●) and Si LRP1-1 treatment (●) obtained with the p-polarized light ($K_D$ values are provided in Supplementary Fig. 2E). **d** Percentage of CXCR3-A at the cell membrane assessed by FACS in Si CTRL- or Si LRP1-1-treated cells. **e** CXCR3 immunostaining in U87-CXCR3-A cells treated with Si CTRL or three different siRNA against human LRP1. Scale: 5 μm. **f** The amount of CXCR3-A in cell membrane extracts assessed by western blotting. Caveolin as a membrane loading control. **g** CXCR3-A ($\lambda = 488$ nm), LRP1 ($\lambda = 547$ nm), and clathrin ($\lambda = 647$ nm) immunostainings in U87-CXCR3-A cells treated with Si CTRL or Si LRP1-1 upon stimulation (PS372424, 100 ng/ml, 60 min). All sections were observed at ×630 magnification under confocal laser scanning microscope (Nikon eclipse Ti). Scale: 5 μm. Results from three independent experiments combined to calculate mean and SEM, and values normalized to those obtained for the control condition. ***$P<0.001$, **$P<0.01$, *$P<0.05$. Multiple comparisons were performed with one-way analysis of variance, followed by Bonferroni post hoc tests

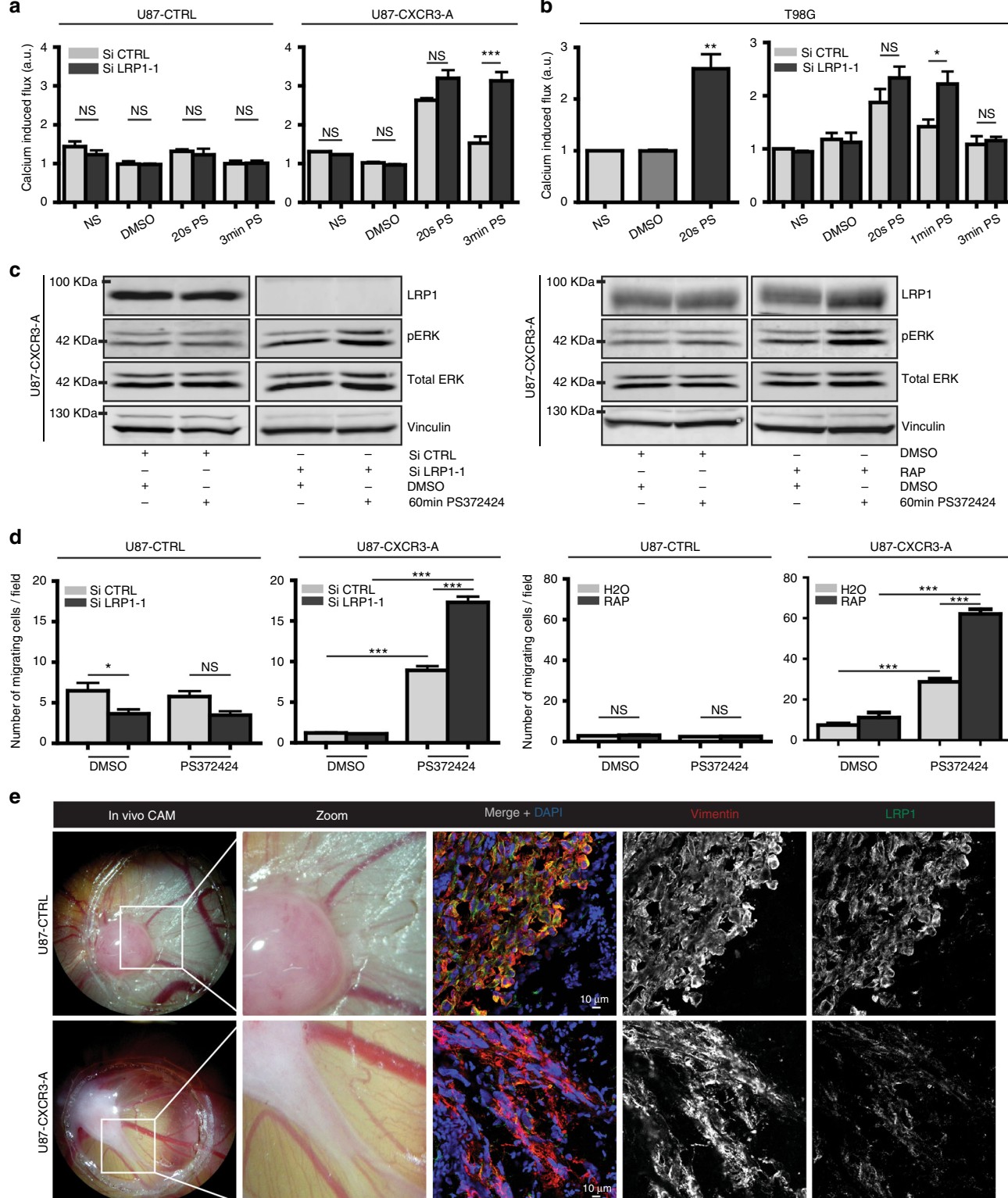

**Fig. 8** Role of LRP1 in CXCR3-A activation and function. **a** Calcium flux assessed by FACS analysis in U87-CTRL and U87-CXCR3-A cells after treatment with control SiRNA (Si CTRL) or LRP1 SiRNA (Si LRP1-1). **b** Calcium flux in T98G cells, which express endogenous level of CXCR3-A. NS non-stimulated condition, DMSO negative control, 20 s PS 20 s post stimulation, 3 min PS 3 min post stimulation, a.u. arbitrary unit. ***$P<0.001$, **$P<0.01$, *$P<0.05$, NS non-significant. **c** Western blot for phosphorylated ERK1/2 in U87-CXCR3-A cell extract from Si LRP1-1 or RAP-treated cells after 60 min of stimulation with PS372424. Vinculin and total ERK used as loading controls. **d** Migration assays ($n = 3$) on U87-CTRL or U87-CXCR3-A cells performed after silencing of LRP1 (left panels) or inhibition of LRP1 (RAP, right panels). ***$P<0.001$, NS non-significant. **e** LRP1 ($\lambda = 488$ nm) and vimentin ($\lambda = 547$ nm) immunostaining were performed at the interface between the CAM and the U87-CTRL or U87-CXCR3-A tumor. Nuclei were stained with DAPI ($\lambda = 405$ nm). Scale: 10 μm. All the results from three independent experiments were combined to calculate mean and SEM, and values were normalized to those obtained for the control condition. Multiple comparisons were performed with one-way analysis of variance, followed by Bonferroni post hoc tests

medium (NBM, Thermo Fisher Scientific) supplemented with 0.2% heparin and 0.02% basic fibroblast growth factors. Adherent human embryonic kidney cell line (HEK-293, ATCC), coronay artery smooth muscle cells (CaSMC, Lonza), and glioma cell lines (U87 (ATCC), U118 (ATCC), 1321N1 (Sigma), NHATS&N-HATSR) (kindly donated by Dr. K. Sasai, Hokkaido University, Sapporo)[64] and T98G (Sigma) were regularly tested for contamination and were all mycoplasma-free. Cells were cultured in Dulbecco's Modified Eagle's medium (DMEM, Thermo

Fisher Scientific) supplemented with 10% fetal bovine serum (FBS), 5% antibiotics (Penicillin and Streptavidin), and 5% L-glutamine. All cells were grown at 37 °C, 5% $CO_2$, and split at 70–90% of confluence with 0.25% Trypsin. Potential mis-sidentified cell lines including HEK-293 and U118 were used for the following reasons: Hek-293 cells do not express CXCR3 (Fig. 1a) and LRP1 (Fig. 4b), and were engineered with CXCR3-A expression vectors; U118 were only used for the analysis of expression of CXCR3, but were not used in any other experiment. All

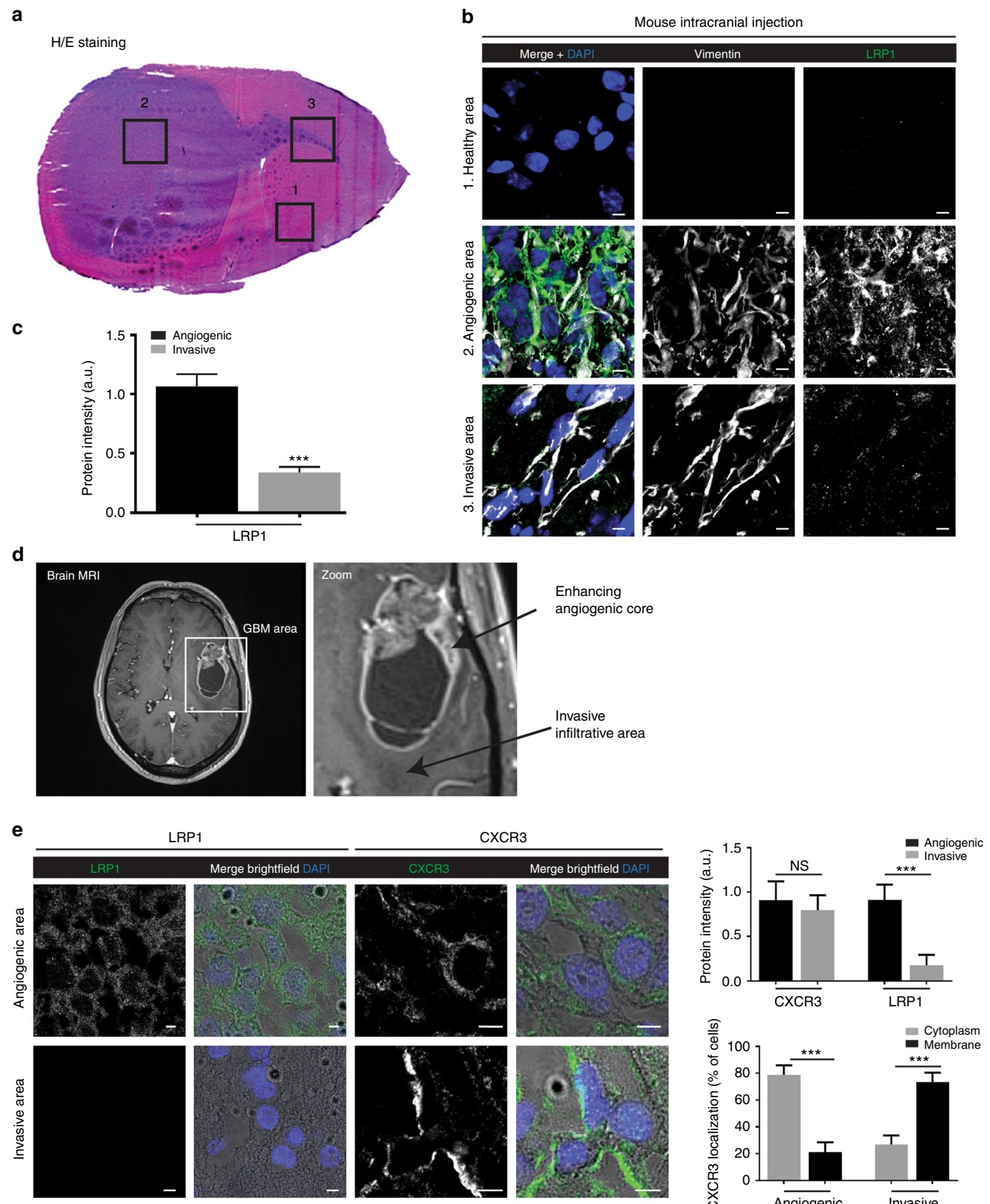

cells used for phenotypic and functional studies have been further characterized more in detail by cGH array (P3) and by cell authentification using Promega Powerplex21 Kit (Eurofins, GE) (cell lines). Cell transfection was performed using Effecten kit (Qiagen) according to the manufacturer's instruction. U87-CTRL (U87 stably transfected with empty pCDNA3.1 vectors) and U87-CXCR3-A cells (U87 stably transfected with pCDNA3.1+CXCR3-A vectors) were grown in the presence of G418 antibiotics (500 μg/ml). HEK-CTRL (HEK-293 stably transfected with empty pEGFPN2 vectors) and HEK-CXCR3-A cells (HEK-293 stably trans-fected with pEGFPN2+CXCR3-A vectors) were grown in the presence of Zeocin (200 μg/ml). A HA-tagged mini-LRP1 construct carrying only the ligand-binding domain IV (HA-LRPIV) was generated as described[33] and used to stably transfect both HEK-CTRL and HEK-CXCR3-A. After transfection, these cells were grown in the presence of Zeocin (200 μg/ml) and G418 (750 μg/ml).

For inhibition experiments, cells were pretreated: 1 h with SCH546738 at 2.2 nM, 15 min with Pitstop2 at 30 μM, or 24 h with RAP at 5 μM.

P3 xenograft spheroids were maintained in culture onto agar-layered flasks and then transplanted into randomly selected immunodeficient RAGγ2C$^{-/-}$. Male RAGγ2C$^{-/-}$ mice ($n = 10$, 7–9-week-old) were housed and treated in the animal facility of Bordeaux University ("Animalerie Mutualisée Bordeaux"). Power calculation was not done in this case since these are qualitative data and each animal is its own control (comparison of central and invasive areas for each animal). All animal procedures have been done according to the institutional guidelines and approved by the local ethics committee (agreement number: C-33-522-2). Animals were killed and tumoral brains were cut into 10 μm cryosections for immunostainings.

**CAM assays**. Fertilized chicken eggs (*Gallus gallus*) (less sentient model according to 3R rule) were incubated at 37 °C and 65% humidified atmosphere and submitted to a rotation of 25° every 4 h during 3 days. On day 3 of development, a window was made in the eggshell and sealed with Durapore tape. On day 10, plastic rings (made from Nunc Thermanox coverslips) were put on the CAM. Three millions of U87-CTRL or U87-CXCR3-A cells were deposited as a thin layer on the surface of the CAM, inside the plastic ring, after gentle laceration of the surface. On day 17, the CAMs were fixed in vivo with 4% paraformaldehyde for 30 min at room temperature. The area containing the ring was cut out and included in optimal cutting temperature compound (OCT) or directly frozen in liquid nitrogen, for further analysis. For the first condition, 40 eggs each were implanted with either U87-CTRL cells or U87-CXCR3-A cells. Among those, 21 U87-CTRL eggs and 23 U87-CXCR3-A eggs survived, analyzed by biomicroscopy and further processed. 1/3 of the CAM were included in OCT for histology and immunostaining and 2/3 were frozen for mRNA and protein extraction. For the second condition, 24 U87-CXCR3-A CAMs were treated with SCH546738 and 10 U87-CXCR3-A CAM with DMSO. Eggs for cell implantation or SCH546738 treatment were randomly selected. Expecting that 75% of the U87-CXCR3-A eggs would have an invasive phenotype, that 75% of the U87-CTRL eggs would have an angiogenic phenotype and similarly that 75% of the untreated (DMSO) U87-CXCR3-A eggs would have an invasive phenotype and that 75% of the treated U87-CXCR3-A eggs would have an angiogenic phenotype, these group sizes correspond, respectively, to a 94% and a 80% power for two sided tests comparing two binomial proportions with significance level 0.05[65].

**Glioblastoma patients**. Patients with preoperative MR imaging studies suggesting a presumptive diagnosis of GBM (defined as intracerebral mass lesion with contrast enhancement area at post gadolinium T1-weighted MR) were included in the study Volumetric FLAIR and post gadolinium T1-weighted images of these patients were obtained in the preoperative period and loaded into the neuronavigation plan (Brainlab workstation) to be available during surgery for intraoperative tracking. Informed written consent was obtained from all subjects (Department of Neuro-surgery, Humanitas, Milano acoording to Humanitas ethical committee regulations).

At surgery, various samples were collected at the peripheral region of the tumor (infiltrative area defined as FLAIR abnormalities around and distant from post gadolinium T1 contrast enhancement ring) or at the tumor core (defined as

contrast enhancement area at the tumor ring in post gadolinium T1-weighted images). The site of collection was registered every time on the neuronavigation system and data were stored and collected for subsequent analysis. Part of the samples were immediatly frozen and stored in the tumor bank. Part of the samples were sent for routine histological and molecular analysis (IDH1 mutation, MGMT promoter methylation status, 1p/19q codeletion, ATRX expression). Only patients with a histological diagnosis of GBM and IDH1 wt profile were included.

**Plasmon waveguide resonance**. PWR measurements were performed in a homemade instrument equiped with a He-Ne laser, with a fixed wavelenght in the visible region ($λ = 632$ nm; Melles Griot), a rotating table allowing the incident angle to be changed by steps ≤ 1 mdeg (thus resolution being on that order; Newport) and a photodiode detector (Hamamatsu) to measure the reflected light as a function of the incident angle. The polarization angle of the incident beam is placed at 45° allowing both *p*- and *s*-polarized spectra to be obtained within the same angular scan. The sensor consisted in a BK-7 prism coated with silver and silica (to support waveguide modes)[29]. All measurements were performed at 22 °C in a temperature controlled room. More details about this technology can be found in Harte et al.[29] publication.

The protocol for adhesion of cell fragments on silica (glass slides or PWR sensor) was adapted from reported work from Vogel and collaborators[66]. Briefly, the silica surface was washed with ethanol and cleaned and activated by plasma cleaner for 2 min (Diener). The silica surfaces are then incubated with a solution of polylysine (0.1 mg/ml) for 40 min following wash with PBS buffer. Cells grown to < 50% confluence are washed with PBS and covered with water to induce osmotic swelling of the cells. Imediately, the glass coverslip of the sensor is placed directly on top of cells. Pressure is applied for about 2 min on the glass slide or prism to induce cell rupture and caption of cell fragments. After that they are removed ripping off cell fragments containing specially the upper membrane. The glass slide or sensor is washed with PBS to remove cell debris and kept with buffer to prevent drying and loss of membrane protein activity. PWR measurements are performed right away. The PWR cell sample (a teflon block with a volume capacity of 250 μl) is placed in contact with the prism and filled with PBS. After cell fragment deposition, there are positive shifts in the resonance minimum position that are correlated with the total mass gain occurring. We have observed spectral shifts that correlate with those observed for the deposition of lipid model membrane. In the case presented here, there are areas of the sensor covered with cell membranes and others uncovered (as observed by confocal microscopy on fluorescently labeled cells (Supplementary Fig. 2A), the PWR signal takes into account both covered and uncovered areas as the laser spot makes about 0.5 mm in diameter. At same time, covered areas include both lipids and proteins, so possess a higher mass than that of a pure lipid membrane. Following stabilization of the signal (no changes in the resonance minimum position with time), the ligand is added in a incremental fashion to this chamber and specral shifts followed with time. The ligand mass itself is small and cannot account for the observed spectral changes that are attributed to the changes in receptor conformation following ligand binding (formation of the receptor/ligand R/L complex). Ligand affinity to the receptor present in the cell membrane fragments is calculated by ploting the shifts in the resonance minimum position (corresponding to R/L complex) as a function of ligand concentration (free ligand, L) and fitting using an hyperbolic function that describes ligand binding to a protein (single site) (Graph Pad Prism).

**Protein structure modeling and molecular docking**. There are lacking of structures of CXCR3-A and LRP1-CR26 and CR27. We used protein fold recog-nition server, Phyre2[67], to build the modeling structures of CXCR3 and CR26, CR27, adopting CXCR4 (PDB code 3ODU) and LRP1 CR17 (PDB code 2KNY) as templates. The molecular docking between CXCR3-A and CR26 (and CR27) was predicted by using HADDOCK2.2 software[36, 37] that implements CNS calcula-tion[68]. We followed the protein–protein docking protocol and used the negatively charged residues distributed on CXCR3 extracellular surface and the positively charged residues on CR26 (/CR27) N-terminal half to be the ambiguous interaction restraints. The final complexes were evaluated by HADDOCK score and structure clusters. The structures occurred in the most favorable cluster and with smaller

**Fig. 9** CXCR3 and LRP1 expression in glioblastoma samples from patients. **a** HE staining was performed in the experimental mouse intracranial P3 GBM model that recapitulates the regional heterogeneity of gliomas. (1) healthy brain, (2) angiogenic part, and (3) invasive part. **b** Immunofluorescence with anti-vimentin (sc-6260 clone V9, Santa Cruz) ($λ = 547$ nm) and anti-LRP1 (antibody 2703-1 for LRP1β, Epitomics) ($λ = 488$ nm) antibodies on tissue from healthy mouse brain, angiogenic, or invasive part of the P3 GBM tumor. Scale: 5 μm. **c** Graph depicts the quantification of LRP1 expression in angiogenic and invasive areas ($n = 30$). ***$P < 0.001$. **d** MRI image from a representative glioblastoma (GBM) patient. Samples were taken intraoperatively with the aid of neuronavigation from the tumor angiogenic core (contrast enhancement ring) and the infiltrative area (T1 hypointense abnormality). **e** Immunofluorescence with anti-CXCR3 (antibody 2ar1, Abcam) and anti-LRP1 antibodies (antibody 2703-1 for LRP1β, Epitomics) ($λ = 488$ nm) on tissue samples from either the angiogenic or invasive part of the tumor. Membrane and/or perinuclear localization of CXCR3 and LRP1 were observed at a ×630 magnification with phase contrast brightfield under a confocal laser scanning microscope (Nikon Eclipse ti). Scale: 5 μm. Graphs (protein intensity, $n = 30$, and CXCR3 localization, $n > 300$) depict the quantification from three different patients. NS non-significant. ***$P < 0.001$. Statistical comparison between two groups was performed using the Mann–Whitney *U*-test. Multiple comparisons were performed with one-way analysis of variance, followed by Bonferroni post hoc tests

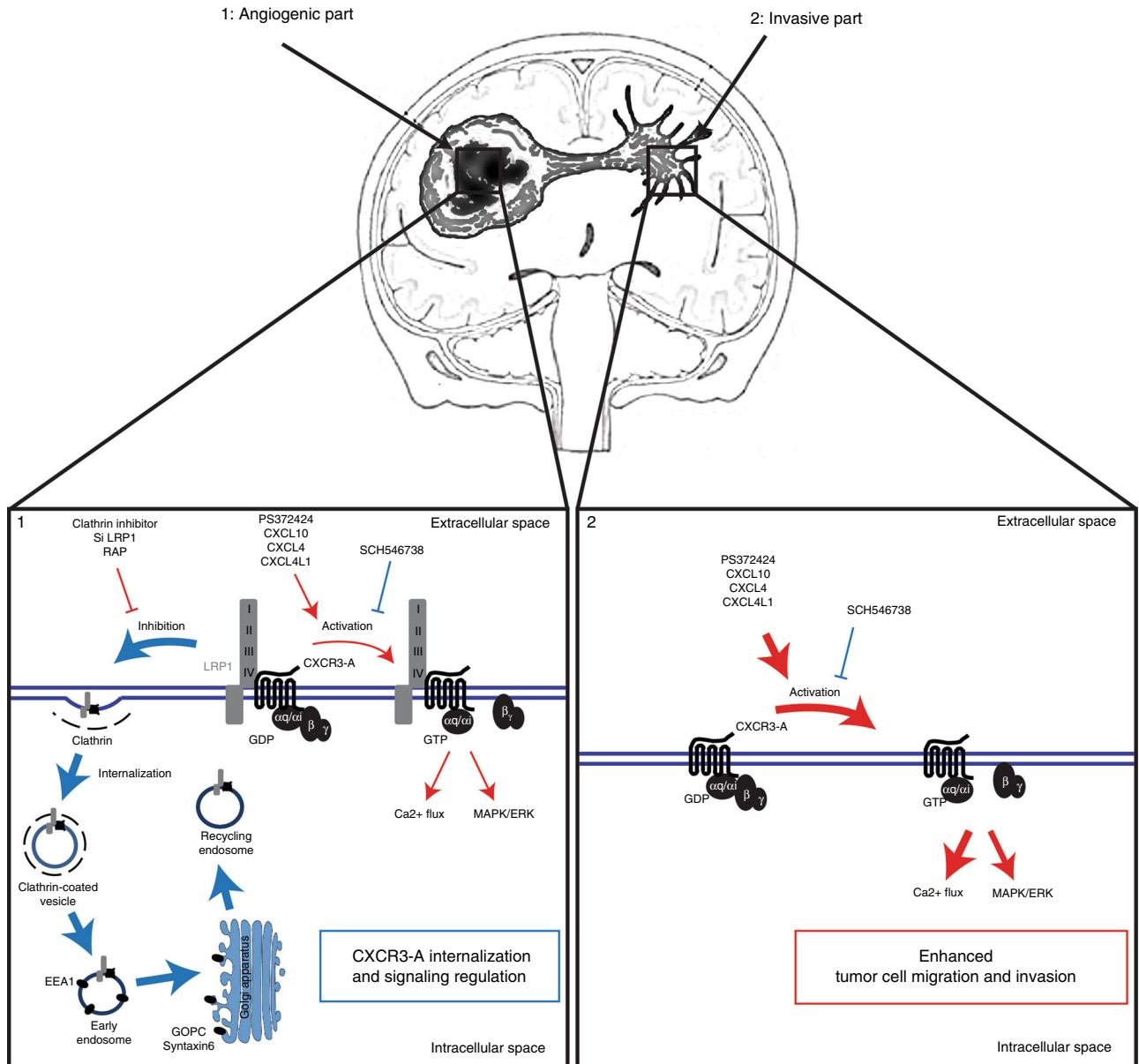

**Fig. 10** Implication of CXCR3-A/LRP1 complex in glioblastoma invasion. Model describing the regulatory role of LRP1 on CXCR3-A function in the angiogenic core of the tumors and in the invasive areas. In contrast to the central core, LRP1 is downregulated in the invasive areas, which leads to an increase in glioblastoma invasion

HADDOCK score were considered to be the most reliable complexes. Meanwhile, because of domain repeats in LRPIV, we selected the CR structures with the N- and C-terminal ends free access for conjugating the previous and next Crs.

For modeling of the CXCR3-A/CXCL10 complex, we used the complex structure of CXCR4 and vMIP-II (pdb code 4RWS)[35] as template. The CXCR4 coordinate was replaced by CXCR3-A and the vMIP-II coordinate by CXCL10. For modeling of the CXCR3-A/PS372424 complex, the binding site of the agonist was derived from the residues identified in the modeling CXCR3–Cp#1 complex[69].

**Western blot and immunoprecipitation analysis.** Cells were washed twice with PBS and dissolved in lysis buffer (10 mM Tris-Hcl, pH 7.4, 150 mM NaCl, 0.5% NP-40, 1% TritonX-100, 1 mM EDTA supplemented with protease and phosphatase inhibitors cocktails (Roche). Protein concentration was quantified by Bradford assay (Euromedex). Cell lysates were resuspended in Laemmli buffer (62.5 mM Tris pH 6.8, 10% glycerol, 2.5% SDS, 2.5% β-mercaptoethanol), boiled for 5 min and resolved by SDS–PAGE. For immunoprecipitation, 500 μg of pre-cleared cell lysate was incubated with 5 μg/ml anti-LRP1 (8G1, ref: ab20384, Abcam), 5 μg/ml anti-CXCR3 antibodies (1C6, ref: ab125255, Abcam), or 5 μg/ml of control IgG (rabbit IgG ref: X0903, DAKO; Mouse IgG ref: MABC002, Millipore) overnight at 4 °C, and then with protein G-Sepharose (Amersham) for 2 h at 4 °C. For both immunoprecipitations, the immunocomplexes were then washed

three times in lysis buffer and twice in PBS. For western blot, 50 μg of protein cell lysate was used for each conditions. All samples were boiled in 4× sample buffer, subjected to electrophoresis and transferred. Proteins were electroblotted onto a polyvinylidene difluoride (PVDF) membrane (ref: 10344661, Fisher Scientific). Membranes were incubated with blocking buffer (ref: 927-40003, EuroBio) for 1 h, probed overnight at 4 °C with the primary antibodies of interest (Anti-LRP1 (ref: 2703-1, dilution: 1/5000), anti-LRP1 (ref: D2642, dilution: 1/1000), anti-CXCR3 (ref: ab77907, dilution: 1/500), anti-CXCR4 (ref: ab72100, dilution: 1/200), anti-CXCR7 (ref: ab124824, dilution: 1/100) PathScan Multiplex western Cocktail I (ref: 5301S, dilution: 1/250), total p42/p44 MAPK (ref: 9102, dilution: 1/1000), anti-vinculin (ref: V9131, dilution: 1/2000), anti-GFP (ref: sc-9996, dilution: 1/2000), anti-clathrin from Vesicle Trafficking Antibody Sampler Kit (ref: 9765, dilution: 1/1000), anti-caveolin1 from Vesicle Trafficking Antibody Sampler Kit (ref: 9765, dilution: 1/1000)), followed by detection using secondary antibodies coupled to Fluoroprobes (goat anti-rabbit IR Dye 680CW, ref: 926-68021; goat anti-rabbit IR Dye 800CW, ref: 926-32211; goat anti-mouse IR Dye 680CW, 926-68020; goat anti-mouse IR Dye 800CW, ref: 926-32210 or goat anti rat IR Dye 800CW, ref: 926-32219, dilution: 1/5000, Li-Cor Biosciences) and Odyssey infrared imaging system (Li-Cor Biosciences, Nebraska, US). Densitometry analysis was performed using Image Studio Lite 4.0 software. Uncropped immunoblots are depicted in Supplementary Figs. 7–11.

**Indirect immunofluorescence staining**. Cells were seeded on glass coverslips and fixed 10 min with 4% paraformaldehyde at room temperature. Cells were permeabilized 10 min with 0.1% TritonX-100, washed with PBS and incubated for 1 h with blocking buffer (PBS containing 2% FBS and 1% BSA) at room temperature. Cells were incubated with primary antibodies diluted in blocking buffer overnight at 4 °C (anti-CXCR3 (ref: ab64714, dilution: 1/300), anti-LRP1 (ref: 2703-1, dilution: 1/500), anti-clathrin from Vesicle Trafficking Antibody Sampler Kit (ref: 9765, dilution: 1/50), anti-caveolin1 from Vesicle Trafficking Antibody Sampler Kit (ref: 9765, dilution: 1/400), anti-EEA1 Vesicle Trafficking Antibody Sampler Kit (ref: 9765, dilution: 1/200), anti-GOPC from Vesicle Trafficking Antibody Sampler Kit (ref: 9765, dilution: 1/100), anti-syntaxin6 from Vesicle Trafficking Antibody Sampler Kit (ref: 9765, dilution: 1/100), anti-Rab5 from Endosomal Marker Antibody Sampler Kit (ref: 12666, dilution: 1/200), anti- β-Arrestin (ref: 610550, dilution: 1/200)), washed with PBS, and incubated with secondary antibodies diluted in blocking buffer (Interchim, FluoProbes 488H goat anti-mouse IgG (H +L) (A11001), FluoProbes 647H donkey anti-mouse IgG (H+L) (FP-SC4110), FluoProbes 488H donkey anti-rabbit IgG (H+L) (FP-SA5110) or FluoProbes 647H goat anti-rabbit IgG (A-21246), dilution: 1/500). Dapi was used as counterstained agent to label nuclei (Fisher Scientific, ref: 10374168, dilution: 1/2000). Coverslips were mounted using Prolong Gold antifade reagent (Fisher Scientific, ref: 11559306). For STED microscopy staining, secondary antibody (STAR 580 and ATTO 647N) were a kind gift from the Bordeaux Imaging Center (BIC).

**Histological and immunofluorescence analyses**. CAM tissues embedded in OCT were cut in frozen sections of 10 μm using the cryostat LEICA CM1900. For histological analyses, frozen sections were stained with haematoxylin and eosin. For immunohistofluorescence analyzes, frozen sections were incubated with primary antibodies anti-vimentin (ref: sc-6260 clone V9, dilution: 1/400) anti-CXCR3 (ref: ab64714, dilution: 1/300), and anti-LRP1 (ref: 2703-1, dilution: 1/500). Fluorescent secondary antibodies (the same as described in indirect immunofluorescence staining) were used for labeling and DAPI was used to stain nuclei (Fisher Scientific, ref: 10374168, dilution: 1/2000).

**Confocal and STED microscopy**. Confocal images were acquired with a Nikon Eclipse Ti inverted laser scanning fluorescence microscope equipped with acquisition software (NIS-Element, NIS) and 63× oil immersion objective (numerical aperture (NA), 1.4). Triple or quadruple imaging were obtained using selective laser excitation (405, 488, 547, or 647 nm). Each channel was imaged sequentially before merging. Z-stack images were taken at sequential 1 μm depth slices and 3D representation were performed on NIS Element software. STED images were acquired in the BIC institute with a Leica SP8-STED microscope equipped with LAS-AF software and with 100× oil immersion objective. The depletion laser was used at 775 nm. Plot profile analysis was performed on ImageJ software.

**Calcium measurement**. Intracellular calcium concentration was measured using Calcium Assay Kit (ref: 640176, BD Pharmingen) according to the manufacturer's procedure. In brief, cells were placed in suspension and ×1 Dye loading solution was added for 20 min at 37 °C. Calcium concentration was then recorded for 5 min on the AccuriC6 flow cytometer. CXCR3 agonist stimulation was performed after 2 min of recording and intracellular calcium increase was visualized. Calcium concentration after stimulation was normalized by calcium concentration before stimulation. Cells without ×1 dye loading solution were used as a negative control in order to quantify the background level.

**Proximity ligation assay**. PLA was performed according to the manufacturer's procedure using the Duolink In Situ Orange Starter Kit Mouse/Rabbit (ref: DUO92102, Sigma). The primary antibodies used were for CXCR3 mouse antibody 2ar1 (ref: ab64714, dilution: 1/300) and for LRP1 rabbit antibody (ref: 2703-1, dilution: 1/500). Before the PLA procedure with secondary antibodies, the protocol described for the indirect immunofluorescence staining was used.

**Immunogold and electron microscopic localization**. Subconfluent U87-CXCR3-A cells stimulated 60 min with CXCR3 agonist were fixed in a mixture of 2% paraformaldehyde and 0.1% glutaraldéhyde in 0.1 M sodium phosphate buffer, pH 7.4 for 1 h and kept in 0.1 M sodium phosphate buffer until further processing. Cell samples were embedded in 10% gelatin, which was solidified on ice. Blocks with cells were immersed in 2.3 M sucrose in phosphate buffer at 4 °C, and ultra-thin cryosections were single or double immunolabeled with 6- and 15-nm protein A-conjugated colloidal gold probes. The rabbit antibodies used were against syntaxin 6 (6 nm, ref: 2869 Cell Signaling, dilution: 1/10) and the mouse antibodies used were agiants CXCR3 (15 nm, ref: 2ar1 ab64714, dilution: 1/300). The labeled sections were contrasted with uranyl acetate and embedded in methyl cellulose.

**Statistical analysis**. Statistical analysis was performed using the GraphPad Prism 5 software and NIS-Element AR 64-Bit software for Pearson's analyses. For the statistical tests used, see figure legends.

**Data availability**. All data are available within the Article and Supplementary Files, or available from the authors upon request.

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

## Acknowledgements

We would like to thank Dr Alexandre Dubrac (Yale University, New Haven, USA) for the kind gift of pEGFPN2-CXCR3-A, Christel Poujol, Melina Patrel, Etienne Gontier, and the Bordeaux Imaging Center (University of Bordeaux, Bordeaux, France) for STED and electronic microscopy acquisition and interpretation of data, Dr K. Sasai, Hokkaido University, Sapporo for NHATS&NHATSR, and Pr. Dudley Strickland (University of Maryland, Baltimore, USA) for the kind gift of goat antibody D2642 against LRP1. We would like to thank Dr Laura Salabert (Inserm U1029, LAMC, Bordeaux, France) for gift and culture of glioma patient-derived cell lines, Mitchell Kramer, and Kenza Rhalies for qPCR experiments on CAM and Christel Herold-Mende for providing NCH421K cells (University of Heidelberg, Germany).

## Author contributions

K.B., N.P., I.D.A., Y.-P.C., T.D., M.C., and C.B. contributed to in vitro and in vivo experiments. I.D.A. contributed to the PWR experiments. T.D. and R.B. to in vitro glioma patient-derived cells generation and P3 xenograft spheroids. L.B. and M.R. contributed to the obtention of glioma patient tissue samples. K.B., N.P., I.D.A., and C.B. contributed to analysis and interpretation of data. Y.-P.C, Y.-Z.L., and S.-C.S. contributed to modeling analysis and results. S.D. contributed to mini-LRP1 domain IV vector generation. S.D., R.B., and S.-C.S. revised the paper critically for important intellectual content. A.B. and C.B. contributed to the design of research, supervised research, and wrote the paper. K.B. and C.B. contributed to the conception of the figures.

## Additional information

**Competing interests:** The authors declare no competing financial interests.

