## [Peer review file · Nature Communications]

Reviewers' comments:

Reviewer #1 (Remarks to the Author):

This paper focuses on the CXCR3-A splice isoform of the chemokine receptor CXCR3. The authors use plasmon waveguide resonance to demonstrate that, "Agonist stimulation induces an anisotropic response with conformational changes of CXCR3-A along its longitudinal axis." The authors go on to demonstrate that CXCR3-A interacted with LRP1, that CXCR3-A is internalized via clathrin-coated vesicles and recycled by retrograde trafficking; and that silencing of LRP1 increased the magnitude of ligand-induced conformational change with CXCR3-A localized at the cell membrane, leading to a sustained receptor activity and an increase in tumor cell migration. These results were validated in patient-derived glioma cells and patient samples. This is a detailed but difficult to read paper. I would encourage the authors to reorganize the data, and simplify the message, and data shown.

The authors analyze expression of CXCR3 in a handful of lines, and based on its low expression, pick U8 cells for future experiments in which they compare cells transduced with CXCR3 with parental cells. Can the authors also provide expression data from public datasets of low and high grade glioma to analyze expression levels. The use of a single cell line is also an issue. Why not knock down or delete CXCR3 from a line with high levels, to confirm key findings made with U87.

Fig 1 claims to analyze gliomas of different grades, but cell lines I think are GBM or immortalized astrocytes. Can authors both clarify and also cite sources and information for cell lines used? How did authors pick dose for CXCR3 antagonist, and what was the approach used to assure that the effect on migration was on-target.

Reviewer #2 (Remarks to the Author):

Boye, et al., LRP1 as a novel regulator of chemokine CXCR3 conformation, activity and trafficking: relevance for tumor cell invasion.

This study begins by cataloguing CXCR3-A and -B isoform expression in various cultured glioma cells. The U87 line was selected for further study due to low expression of CXCR3-A and -B. Forced expression of CXCR3-A altered cells to become more motile and invasive, in an agonist-dependent manner. Binding of CXCR3-A to ligand was detected and quantified by PWR on adherent cell membrane fragments. Agonist application triggered Ca²⁺ and MAPK signaling in intact CXCR3-A-expressing cells. LRP1 co-localized and co-IPed with CXCR3-A. Further, LRP1-CXCR3 binding was inhibited by RAP and involved the LRP1 extracellular domain IV. CXCR3-A could be docked computationally onto LRP1 repeat CR26. Cell co-localization studies showed CXCR3-A and LRP1 in a predominantly juxtannuclear position at steady state. Both receptors appeared to move to the PM with ligand stimulation, and then return to the intracellular pool over the course of about 1 h. The latter was inhibited by the Pitstop 2 inhibitor. Returning to PWR, LRP1 RNAi altered the CXCR3-A conformational changes measured and increased CXCR3 number at the PM without affecting the KD. LRP1 RNAi also increased CXCR3 at PM followed by IF. Interfering with LRP1 extended the duration of Ca²⁺ and MAPK signaling in glioma tumor cells. Loss of LRP1 also led to increased cell motility and invasion. Finally, in human high grade glioblastoma tissue sections, CXCR-3 was elevated and LRP1 reduced in the invasive portion of the tumor. The authors propose a working model in which LRP1 negatively regulates CXCR3 signal intensity and duration and, in glioma, decreased LRP1 causes excessive signaling and tumor invasion.

This body of work represents a large array of experimental approaches and analysis techniques. The role of CXCR3-A in glioma spread could be clinically important. The possible involvement of

LRP1 in CXCR3-A signaling is interesting. Yet, despite the amount of work and potential significance, there are a number of substantial shortcomings and limitations to the study. Overall, this makes the study more correlative than definitive in showing causation in this reviewer's view.

Significantly, they have not performed the reciprocal experiment where they explore whether increased expression of LRP1 diminishes ligand-triggered CXCR3-A-expressing U87 cell migration and invasion. Moreover, given the level of expression of LRP1 in the U87 cells (Fig. 3 and 5), it is not obvious why this is not sufficient to suppress over expression of CXCR3-A in these cells.

This highlights a crucial problem with the manuscript: Much of the study boils down to relative stoichiometries of the two transmembrane components, CXCR3-A and LRP1. It seems that in general, LRP1 will be present in excess. However, actual amounts are not compared at all in a biochemically-meaningful way. Binding studies with labelled CXCR3 and LRP1 ligands could give accurate numbers for the relative copy number in the cells they are investigating. At a minimal level, I do not believe that the reader can compare directly the level of CXCR3-A expression in the constructed/transfected U87 cells with the high expressing T98G cells.

Another confounding issue is that the final clinical significance shown in Fig. 7 is interpreted as "...LRP1 expression is totally shut-down in invasive zones of the tumor." They also assert: "The extinction of LRP1 in invasive tumor areas ..." This strong claim can only be made after being validated with antibodies directed against other different regions of this large transmembrane receptor, as the proteolytic environment of the invasive tumor region is likely to be different to the angiogenic part. In this regard, several antibodies to LRP1 are listed in the Methods section, but the reader is unable to identify which was used for what experiments.

The computational docking model in Fig. 3G seems to indicate that CR26 would engage CXCR3 at or near the extracellular GPCR ligand binding interface? Is this compatible with classical full-length chemokine binding simultaneously? Is it possible that LRP1 affects the dissociation rate constant of the (small) ligand to impact signal strength? Needless to say, it is important to definitively establish whether the effect of LRP1 is restricted to the agonist PS372424, or also occurs with full-length peptide ligands.

The ability of ligand to quickly alter the distribution of both CXCR3-A and LRP1 from endosomes to the PM seems odd. LRP1 is considered to be a constitutively internalized nutrient-type receptor and has multiple sorting signals that make LRP1 uptake more rapid than the LDLR, VLDL or ApoER2, for example. So why would ligand retard this internalization? The authors make no attempt to rationalize this for the reader. Are they arguing that ALL of the cellular LRP is associated with CXCR3-A?

One conclusion drawn from Fig. 5 is that LRP1 RNAi "... was associated with impaired clathrin-coated vesicle formation (Fig. 5G)." There is little accepted evidence that LRP1 positively regulates CCV formation generally in cultured cells. If the authors have additional supportive evidence from the literature this should be explicitly cited. Moreover, if correct, it would attest to the high abundance of LRP1 in the U87 cell line.

The PWR data are opaque and not presented in a coherent mechanistic manner. Given they are looking at small pieces of poorly characterized adherent membrane fragments, in the absence of any cytosolic support or source of cofactors, it is unclear what exactly it is they are

reporting. The non-specialist reader should be considered when drafting these results and the accompanying discussion.

In Fig. 5E and G, it appears that different focal planes are being compared. In E, Si CTRL, looks like a medial optical section, while the SI LRP1 panels are ventral/basal sections to see PM/lamellapodia. This seems also the case in G.

Are the authors arguing that beta-arrestins are not important for CXCR3 internalization and/or post-uptake signaling? There is an odd reference to some beta-arrestin data on page 15, but I do not see this referred to directly in the main body of the Results section. Any interplay between the beta-arrestins and LRP for clathrin-coated vesicle sorting of CXCR3 is left to the reader to interpret alone.

Another issues is that the authors CANNOT claim that Pitstop 2 is a specific pharmacologic inhibitor of clathrin-mediated endocytosis. This compound has been thoroughly discredited and stops essentially all trafficking from the PM and also affects nuclear pore trafficking. This is published.

Reviewer #3 (Remarks to the Author):

The manuscript submitted by Boye and coworkers describes nicely the colocalisation of CXCR3 and LRP1 in glioma cell lines and the CXCR3 recycling pathway. They showed that LRP1 is a novel regulator of CXCR3 mechanism of action. Overall, the study is well-designed and combined multidisciplinary techniques.

I suggest acceptance of this manuscript after revision.

MAJOR COMMENTS

1. In Figure 1A, authors described the CXCR3 mRNA copy number in several cell lines. They indicate, on one hand, that T98G expressed significantly more CXCR3A, and on the other hand, that U87 expressed significantly less CXCR3B. But compare to what?
2. In figure 1B, authors showed a western blot for total CXCR3 protein. The abundance of the protein did not fit with the relative gene expression showed in figure 1A. Is it due to the fact that western blot evaluate CXCR3 A+B?
3. In the same figure, authors are using vinculin as a loading control. As vinculin is a protein associated with focal adhesion plaques. Is it the relevant loading control, as CXCR3 seemed to be involved in invasion?
4. Authors showed nicely on CAM, that CXCR3A expressing U87 cells were more invasive than their control counterpart. Did they observe the same phenomenon with 1321N1 or NHATS cells expressing naturally more CXCR3?
5. Authors showed a close-colocalization of CXCR3 and LRP1 using STED HR microscope. They stated that the proteins were colocalized in the cytoplasm, near the nucleus (line 158). It's not obvious as no nuclear staining were used as a topological help. I strongly advise authors to include such a staining.
6. I advice authors to confirm their colocalization results (CXCR3-LRP1) by using Proximity Ligation Assay.
7. I strongly suggest validating the localization of CXCR3 in the recycling compartment by immunogold techniques or correlative microscopy.
8. One-way ANOVA followed with multiple comparisons were adequately used. However, I strongly suggest the use of Bonferroni post-test instead of Tukey.

MINOR COMMENTS

1. In Figure 1A, authors described the CXCR3 mRNA copy number in several cell lines. I am however disturbed by the use of arbitrary unit. I guess it is a ratio between CXCR3 gene expression and the reference gene expression. In that case, CXCR3 copy number should be replaced by a relative gene expression.
2. Boyé et al stably transfected U87 cells with GFP-CXCR3-A encoding vectors. They showed by western blot the expression of this fusion-protein using GFP antibody. Results will be more convincing by using anti-CXCR3 antibody.
3. In the Figure S1A, authors indicated that CXCR3B is not expressed in U87 after CXCR3-A transfection. They should replace "NS" (not significant) by "ND" (not detected).
4. In the Figure S1B, they showed the CXCR3 protein abundance after siRNA transfection. CXCR3-9 siRNA worked pretty well (90% depletion) but it was not the case of the 3 others siRNA. Do the authors have a comment on this result? Statistically, I'm surprised that CXCR3-9 siRNA depletion was less significant than the other depletion.
5. In figure S1C legend, they should replace "pv21" by "pV21".
6. In figure S1D legend, they should replace "Western" by "western" as western, at the opposite of Southern, is not a family name. Same comment for lines 106 and 241.
7. Authors used the CAM model to generate U87 tumors expressing CXCR3. Ideally they should prove that CXCR3 antibody does not cross react with a chicken protein. Did they try to grow HEK tumors on CAM and observe the same result?
8. Figure 2C should have the same x-axis labeling.
9. Authors stated that CXCR3 ligands (PS372424, CXCL10 & CXCL4L1) increased Erk phosphorylation in their CXCR3-expressing U87. Their results are obvious but will be more convincing by adding quantification.
10. CXCR3 and LRP1 were partially colocalized by STED HR microscopy using metamorph software (line 160). Authors should indicate what was the procedure used: Pearson, Mander, Li, ...
11. Authors used RAP in order to antagonize LRP1 ligand binding. They observed by coIP a decrease of the CXCR3:LRP1 complex formation. I notice a decrease of LRP1beta abundance in the complex. This decrease seemed proportional to the CXCR3 decrease. Do the authors have considered a specific interaction of CXCR3 and LRP1beta?
12. Authors should replace "Down-regulation of LRP-1 expression" (line 225) by "LRP-1 depletion".
13. In figure 5A, authors showed the result of the LRP1 siRNA transfection in U87 cells. The U87-CTRL western-blot does not fit (see siRNA LRP1-1) with the quantification presented in the graphic.
14. Line 230, authors wrote: higher shifts in p- than s- polarisation are observed. IN the corresponding figure (5B), they should use the same y-axis range in order to allow the comparison. I did not understand the link between the Figure 5B and the table below.
15. In all their figures, authors used personalized symbols to indicate statistical significance. It is more pertinent to use: one star for $p < 0.05$, two stars for $p < 0.01$ and 3 stars for $p < 0.001$.
16. In figure 6E the scale bar represents $10\mu\text{m}$ when the legend indicates $5\mu\text{m}$.

Comments by Reviewer 1:

1- This is a detailed but difficult to read paper. I would encourage the authors to reorganize the data, and simplify the message, and data shown.

We have reorganized parts of the manuscript to make the flow of the article more coherent. The result section of the manuscript has now the following structure:

Expression and function of CXCR3 in glioma cells- Conformational and dynamic properties of CXCR3-A - Downstream activation of CXCR3-A in glioma cells - First evidence for an interaction between CXCR3 and LRP1 - CXCR3-A binding to LRP1 through its ligand-binding domain IV - Clathrin-dependent internalization of CXCR3 and LRP1 and recycling - LRP1 as a novel regulator of CXCR3 activation and function - LRP1 and CXCR3 expression in glioma samples from patients.

As it can be seen, we have simplified the structure by fusing paragraphs and rewrote parts of the article in the results section in a more concise manner. Furthermore, we have added new figures and data requested by the reviewers, which also clarifies the message. The manuscript in its revised form seems to us now appropriate for the message we aim to convey.

2- Can the authors also provide expression data from public datasets of low and high grade glioma to analyze expression levels of CXCR3?

To investigate the expression of CXCR3 in glioma, we analyzed bioinformatics data from the Cancer Genome Atlas (TCGA) gene expression profiling database. These data are presented in the Figure 1 of the letter. The most complete Cancer Genome Atlas (CGA) dataset for grade II and III glioma is the RNA Sequencing data. Figure 1A shows CXCR3 mRNA levels in lower grade gliomas. CXCR3 expression is predictive of survival in low grade glioma when the patients are split into low, medium and high expression groups. Figure 1B shows that CXCR3 expression increased from grade II to grade III and is predictive of survival when patients are split into low and high expression groups. Using the Sun et al. gene expression dataset composed of grade II, III and IV gliomas, this confirms the relative expression of CXCR3 in all 3 grades (Fig. 1C). These bioinformatic data are in agreement with several reports on the expression of CXCR3 in glioma. CXCR3 has been reported to be overexpressed in glioblastoma and astrocytoma. The ligand/receptor pair CXCL10/CXCR3 also showed a strong correlation with tumor grade¹⁻³. One must keep in mind that TCGA data are collected from the core of the tumor and do not include various tumor areas such as the infiltrative/invasive part of the tumor. In our study, multiple samples per patient were taken from central and peripheral tumor areas. It is to emphasize that our study focuses on the differential intracellular distribution of CXCR3 and not on overall expression levels measured by transcriptomics which we did not expect to change. Overall expression levels are only of concern for LRP1, because its expression is down-regulated in the infiltrative area which is in agreement with the general contention of our study.

Figure 1: TCGA bioinformatical data of CXCR3 expression in gliomas.

3- The use of a single cell line is also an issue.

We agree that the use of a single cell line is an issue. Therefore, a control cell line (HEK-293 transfected with CXCR3-A (Fig. S1 of the article) and LRP1 (Fig. 4B of the article), and another glioblastoma cell line that expresses endogeneously CXCR3-A (T98G, Figs. 1A and 1B of the article) were also used in our study. These cells were used for activation studies of CXCR3 after stimulation and interaction studies between CXCR3 and LRP1 and for the elucidation of the role of LRP1 (biological function and internalization of CXCR3; Figs. 2-D,-E,-F,-H, 5-B, 6-E, 6-F, 8-B and Figs. S2-A, S3, S4 and S5 of the article). In order to make our findings more clinically relevant, a patient derived-xenograft model (P3, Figure 9-A,-B and -C of the article) and glioblastoma patient's tissue (Figure 9-D and -E of the article) were also used. In this case, the status of CXCR3 and LRP1 in both the central and the peripheral tumor areas was examined.

4- Why not knock down or delete CXCR3 from a line with high levels, to confirm key findings made with U87.

According to the reviewer's suggestions, we examined the effect of CXCR3 silencing in another glioma cell line in order to confirm our findings. Therefore, further experiments have been carried out in the T98G cell line that highly expresses CXCR3-A mRNA (Fig. 1A of the article) and CXCR3 protein (Fig. 1B of the manuscript). Migration assays with the Incucyte live cells analysis system (Essen Bioscience) have been performed which demonstrated that, upon stimulation with the CXCR3 agonist, T98G cell migration was decreased in T98G-CXCR3 depleted cells. These data are presented in Figure 2 of the letter and in Figure S2A of the article.

Figure 2: *In vitro* effect of CXCR3 silencing. Cell migration assay of T98G cells silenced for CXCR3 (Si CXCR3) versus control Si RNA (Si CTRL) stimulated with the agonist of CXCR3 (PS372424, Ago).

5- Can authors both clarify and also cite sources and information for cell lines used?

Regarding the glioma cell lines used in the paper, a table with all the information and sources⁴⁻⁷ has been made below (Table 1 of the letter) and added in Figure S1A of the article.

Cell line	Grade	Histology	Reference
1321N1	II/III	Astrocytoma	Ponten J. et al (4)
NHATS	II/III	IHA	Sasai K. et al. (5)
NHATSR	II/III	IHA	Sasai K. et al. (5)
U87	IV	Glioblastoma	Ponten J. et al.; Allen M. et al. (4, 6)
U118	IV	Glioblastoma	Ponten J. et al. (4)
T98G	IV	Glioblastoma	Stein GH. (7)

Table 1 : Cell types used in the article

6- How did authors pick dose for CXCR3 antagonist?

The CXCR3 antagonist, SCH546738, was synthesized and provided by Chinapeptide, and independently tested for CXCR3 selectivity. The specificity of SCH546738 has been demonstrated and published in the paper of Jenh et al.⁸. They showed that SCH546738 displaced radiolabeled CXCL10 from human CXCR3 with a concentration range from 0.8 to 2.2 nM. We decided to use the calcium flux assays and performed a dose response assay with SCH546738 to confirm these data. The data in Figure 3 of the letter depict a dose response curve using varying concentrations of SCH546738 (from 0.5 nM to 2.2 nM). To fully inhibit CXCR3 in our experiments, we decided to use the maximal dose of 2.2 nM, which is also recommended in the paper of Jenh et al.

Figure 3: Dose-response for SCH546738 CXCR3 antagonist in HEK-CXCR3-A cells.

7- What was the approach used to assure that the effect on migration was on-target?

In the paper, we already showed that U87-CXCR3-A cell migration induced by the CXCR3 agonist is inhibited when cells were pre-treated with the CXCR3 antagonist SCH546738 (Fig. 1C of the article). This is also the case for T98G cells which are silenced for CXCR3 (Fig. S2A of the article). To demonstrate that cell migration is due to the activation of CXCR3-A, we have also further validated the effect of SCH546738 in the chick embryo's chorioallantoic membrane (CAM) model. These data are presented in the Figure 4 of the letter and in Figure S2B of the article. CAM were implanted with U87-CXCR3-A cells and treated with DMSO (control condition) or with the CXCR3 antagonist SCH546738 (10 µg at day 2, 3 and 4 after implantation). U87-CXCR3-A tumors treated with DMSO or with the SCH546738 express the same level of CXCR3-A *in vivo* as measured by qPCR (Fig. 4 of the letter and Fig. S2B of the article). Images show that U87-CXCR3-A tumors treated with SCH546738 are better circumscribed and larger compare to U87-CXCR3-A tumors treated with DMSO and displayed significant reduction of tumor cell invasion. Invasive strands made of tumor cells are barely detected in SCH546738-treated CXCR3-A expressing tumors. These data confirm that targeting CXCR3 *in vivo* inhibits the invasive phenotype observed in U87-CXCR3-A expressing tumors.

We also used another cancer cell type, the pancreatic MIA PaCa-2 cells expressing CXCR3-A (MIA PaCa-2-CXCR3). These cells were cultured with CXCL4L1-producing Hek-293 cells, a ligand for CXCR3, in chronic stimulatory conditions. We verified that this effect is dependent on CXCR3 by using the CXCR3 antagonist SCH546738 and demonstrated that CXCR3 dependent-biological activity is inhibited in the presence of SCH546738⁹. Thus, the specificity of SCH546738 is verified *in vitro* and *in vivo* and the effect is on-target.

Figure 4: *In vivo* effect of CXCR3 inhibition. CAM assays were performed using U87-CXCR3-A cells treated with DMSO (negative control) or CXCR3 antagonist SCH546738 (n= 50 eggs for each condition). Seven days after implantation, relative CXCR3-A mRNA expression in U87-CXCR3-A tumors treated with DMSO or SCH546738 and images of CAM tumors were performed.

Comments by Reviewer 2:

1- Significantly, they have not performed the reciprocal experiment where they explore whether increased expression of LRP1 diminishes ligand-triggered CXCR3-A-expressing U87 cell migration and invasion.

We appreciate the relevance of these comments and have therefore overexpressed LRP1 in the U87-CXCR3-A cells. Full-length LRP1 is nearly impossible to transfect into cells and HA-tagged minireceptor derived from full length LRP-1 carrying only the ligand binding-domain IV (HA-LRPIV) was used instead¹⁰, since CXCR3-A interacts with the fourth ligand binding domain of LRP-1. U87-CXCR3-A cells were transiently transfected with functional HA-LRPIV. Experiments to analyse the effect of HA-LRPIV on U87-CXCR3-A cell migration have been carried out using the Incucyte live cell analysis system (Essen Bioscience). These data are represented in the Figure 5 of the letter. Migration was not modified in U87-CXCR3-A expressing HA-LRPIV cells in comparison to control U87-CXCR3-A cells. We therefore analysed the status of CXCR3 at the cell membrane using FACS analysis. U87-CXCR3-A cells that overexpressed HA-LRPIV did not exhibit modifications in CXCR3 distribution at the cell membrane (Fig. 5-B of the letter). Thus, it seems that mini-receptor of LRP1 is not able to modify CXCR3 distribution when overexpressed in U87 cells.

This may be due to the following reasons:

1/ U87-CXCR3-A express already endogenously high levels of LRP1 (see answer to question 3). Bu G. et al. already reported that U87 cells express the highest amount of LRP1 that makes this cell line an excellent model to study the cellular biology of LRP1¹¹. Further increase of LRP1 will not contribute to increase CXCR3 internalization since already saturating levels are reached.

2/ the complex between CXCR3 and HA-LRPIV is not sufficiently stable and requires full length of LRP1 for stabilization. It is possible that the interaction of CXCR3 with the fourth ligand binding domain of LRP1 needs to be further stabilized by the need of additional factors that are recruited by full length LRP1.

3/ the HA-LRPIV construct is instable and cleaved when expressed in cells. We observed by Western blot a fragment of approximately 100 kDa which corresponds to a cleaved form of LRP1. This is observed after transient and stable transfection (which is more pronounced in the latter case, Figure 5-C of the letter).

Figure 5: Overexpression of the mini-receptors HA-LRP1V.

A: cell migration, B: FACS analysis for CXCR3 et the cell membrane and C: western blot anti-HA for minireceptor HA-LRP1V. * cleaved HA-LRP1V.

2- Moreover, given the level of expression of LRP1 in the U87 cells (Fig. 3 and 5), it is not obvious why this is not sufficient to suppress over expression of CXCR3-A in these cells.

One of our main findings is that LRP1 is a new partner of CXCR3-A involved in its endocytosis process. LRP1 does not suppress overexpression of CXCR3-A but regulates its conformation, activation and trafficking. In fact, it is known that LRP1 is highly express in the brain and in glioma^{11,12}. Even if LRP1 is highly expressed in U87 cells, only a fraction is localized at the plasma membrane, while the majority is found in the intracellular

compartment¹¹. Furthermore, using Stimulated Emission Depletion (STED) microscopy, we have shown that not all of the LRP1 but only 55 % of it is colocalized with CXCR3-A in U87-CXCR3-A cells (Fig. 3A, right panel of the manuscript). Thus, even if endogenous LRP1 expression level is high in our cells, it is not sufficient to “fully” suppress the activation of CXCR3-A. In addition, 30 % of CXCR3 is colocalized with LRP1 which indicates that enough unbound CXCR3-A is present for ligand interaction (Figure 6 of the letter, Figure 3A of the article). Taken together, these findings explain why CXCR3-A overexpression is not suppressed by endogenous LRP1 in U87 cells.

Figure 6: Colocalization with LRP1 by STED in U87-CXCR3-A cells.
*** p<0.001

3- Much of the study boils down to relative stoichiometries of the two transmembrane components, CXCR3-A and LRP1. It seems that in general, LRP1 will be present in excess. However, actual amounts are not compared at all in a biochemically-meaningful way. Binding studies with labelled CXCR3 and LRP1 ligands could give accurate numbers for the relative copy number in the cells they are investigating. At a minimal level, I do not believe that the reader can compare directly the level of CXCR3-A expression in the constructed/transfected U87 cells with the high expressing T98G cells.

We agree that relative amounts of both CXCR3 and LRP1 have not been mentioned. We initially considered mRNA levels only for CXCR3 isoforms using the relative quantification and western blot in our different cell lines (Figs. 1A and 1B of the article).

To address the concern raised by the reviewer, we performed absolute quantification by qPCR and FACS analysis for CXCR3-A and LRP1 in U87-CXCR3-A and T98G cells instead of ligand binding assays. qPCR gives absolute numbers for CXCR3 and LRP1, and FACS analysis estimates the content of both molecules at the cell membrane in a quantitative meaningful way. These data are represented in Figure 7 of the letter. CXCR3 and LRP1 levels measured by qPCR are much higher in U87-CXCR3A cells than in T98G cells (Fig. 7A). To quantify the levels present at the cell membrane, we measured the ratio of CXCR3 and LRP1 by FACS analysis in non-permeabilized cells (Fig. 7B). This did show only a slight difference between U87-CXCR3-A and T98G cells (0.78 in U87-CXCR3A versus 1.06 in T98G). This indicates that, despite the differences in expression in both cell types, the ratio at the cell membrane between CXCR3 and LRP1 was similar. This allows the comparison in terms of CXCR3-dependent biological activity.

Figure 7: Expression of CXCR3 and LRP1 in U87-CXCR3-A and T98G

A: Copy numbers of CXCR3-A and LRP1 mRNA transcripts. B: FACS analyses for CXCR3 and LRP1.

** p<0.01 and *** p<0.001

4- Another confounding issue is that the final clinical significance shown in Fig. 7 is interpreted as “...LRP1 expression is totally shut-down in invasive zones of the tumor.” They also assert: “The extinction of LRP1 in invasive tumor areas ...” This strong claim can only be made after being validated with antibodies directed against other different regions of this large transmembrane receptor, as the proteolytic environment of the invasive tumor region is likely to be different to the angiogenic part. In this regard, several antibodies to LRP1 are listed in the Methods section, but the reader is unable to identify which was used for what experiments.

We agree that the angiogenic core and invasive peripheral tumor areas have microenvironments with different properties due to the heterogeneity of the cells and of the factors they are producing such as proteases, growth factors etc... We have conducted a proteomic study to gain insights into tumor heterogeneity using an experimental intracranial GBM mouse model that recapitulates the regional heterogeneity of gliomas, where human-derived patient spheroids are used for implantation (the P3 model). The P3 model has been extensively characterized by a collaborator of this study¹³. These data are presented in the Figure 8 of the letter. The proteomic data represent the ratio (R) of protein expression between the invasive part (labelled 3 in the Fig. 9A of the article) and the angiogenic part (labelled 2 in the Fig. 9A of the article) in the P3 model (n=4 brains). A ratio ≤ 0.5 corresponds to the downregulated proteins in the invasive part whereas a ratio ≥ 2 corresponds to upregulated proteins in the invasive part. We show that LRP1 is amongst one of the main proteins downregulated in the infiltrative/invasive part in the P3 model. Details regarding the LRP1 antibodies used in this study have been added in the Materials and methods section and in the legend of each figure.

Accession	Description	Symbol	Mediane R	SD	T-Test
P09211	Glutathione S-transferase P OS=Homo sapiens GN=GSTP1 PE=1 SV=2	GSTP1	0,32	0,18	0,01
Q07954	Prolow-density lipoprotein receptor-related protein 1 OS=Homo sapiens	LRP1	0,45	0,22	0,01
P10809	60 kDa heat shock protein, mitochondrial OS=Homo sapiens GN=HSPD1 PE=1 SV=1	HSPD1	0,66	0,19	0,04
Q13885	Tubulin beta-2A chain OS=Homo sapiens GN=TUBB2A PE=1 SV=1	TUBB2A	1,59	0,23	0,01
Q14194	Dihydropyrimidinase-related protein 1 OS=Homo sapiens GN=CRMP1 PE=1 SV=1	CRMP1	2,69	0,81	0,02
P60201	Myelin proteolipid protein OS=Homo sapiens GN=PLP1 PE=1 SV=2	PLP1	7,24	0,36	0,00

Table 2: Table of some of the proteomic data obtained in the P3 model.

Mediane R: mediane ratio of protein expression of the invasive part/angiogenic part (n=4 brains). SD: Standard Deviation.

5- The computational docking model in Fig. 3G seems to indicate that CR26 would engage CXCR3 at or near the extracellular GPCR ligand binding interface? Is this compatible with classical full-length chemokine binding simultaneously? Is it possible that LRP1 affects the dissociation rate constant of the (small) ligand to impact signal strength? Needless to say, it is important to definitively establish whether the effect of LRP1 is restricted to the agonist PS372424, or also occurs with full-length peptide ligands.

Next, we modelled the interaction in the presence of the agonist PS372424 or full length CXCL10 (Figure 8 of the letter, Fig.4D, 4E of the article). Based on the model, CR26 engages CXCR3 mainly through the interaction with ECL2. In presence of PS372424, CR26 binding to CXCR3 is not overlapping with agonist binding to CXCR3, since PS372424 occupies only the inside of the GPCR hydrophobic pocket. Furthermore, CR26 is unable to penetrate deep into the binding pocket to interfere with agonist binding to CXCR3. Comparing the binding mode with that of the agonist, CXCL10 also displays a deep penetration into CXCR3 hydrophobic pocket. Furthermore, CXCL10 has a much wider binding surface and possesses

more interactions with CXCR3, including an interaction with the GPCR receptor's N-terminus and the hydrophobic pocket. Therefore, we expect higher affinity of the full-length chemokine for CXCR3.

The complex model of CXCR3A and its ligands.

Figure 8: Complex model of CXCR3-A and its ligands. A: full-length CXCL10/IP10. B: Agonist PS372424

To further validate these models and to show that the effect of LRP1 is not restricted to the agonist PS372424, PWR experiments were performed in U87-CXCR3-A cells silenced for LRP1 and stimulated with incremental concentrations of the full-length peptide CXCL10 until the saturation of the ligand-binding response. Our data presented in Figure 9 of the letter and in Figure S6A of the article show a strong affinity of CXCL10 for CXCR3-A (K_D of 2.9 \pm 0.8 nM for CXCL10 versus K_D of 36 \pm 9 nM for PS372424; Fig.S6A) and confirms that the effect of the downregulation of LRP1 is not restricted to the small molecular-size agonist PS372424 but also includes the full-length CXCL10 ligand.

Indeed, as with PS372424, binding of CXCL10 to siRNA LRP1 treated cell fragments leads to spectral changes that were about two-fold higher in magnitude than those observed with control siRNA. This is correlated with the higher amount of receptors present in the cell fragments of siRNA treated cells.

Figure 9: Binding of CXCL10 to U87-CXCR3-A cells – effect of siRNA LRP1 treatment. U87 cell membrane fragments expressing the CXCR3-A were treated with siRNA Ctrl and siRNA LRP1. Resonance shifts observed and K_D values: PWR spectral changes induced by CXCL10 binding to the CXCR3-A and dissociation constants.

6- The ability of ligand to quickly alter the distribution of both CXCR3-A and LRP1 from endosomes to the PM seems odd. LRP1 is considered to be a constitutively internalized nutrient-type receptor and has multiple sorting signals that make LRP1 uptake more rapid than the LDLR, VLDL or ApoER2, for example. So why would ligand retard this internalization? The authors make no attempt to rationalize this for the reader. Are they arguing that ALL of the cellular LRP is associated with CXCR3-A?

We clearly show using STED imaging that not all of the cellular LRP1 is associated with CXCR3-A. Indeed 55% of LRP1 is colocalized with CXCR3 (Fig. 3A of the manuscript) and 30 % of CXCR3 with LRP1. Furthermore, the ligand does not need to retard internalization and this was not stated in the manuscript. Once it is bound, the receptor may be activated and this leads to signaling. What we claim is that when LRP1 is down-regulated as observed in the invasive borders, the amount of CXCR3 that stay at the cell membrane is increased and thus more receptor is activated.

7- One conclusion drawn from Fig. 5 is that LRP1 RNAi “... was associated with impaired clathrin-coated vesicle formation (Fig. 5G).” There is little accepted evidence that LRP1 positively regulates CCV formation generally in cultured cells. If the authors have additional supportive evidence from the literature this should be explicitly cited. Moreover, if correct, it would attest to the high abundance of LRP1 in the U87 cell line.

We agree with the reviewer that the link between LRP1 and clathrin has not be explicitly explained albeit there is literature that provide clear evidence. Bu G. et al.¹¹ already attested the high abundance of LRP1 in the U87 cell line and claimed that this is an excellent model to dissect LRP1 cellular biology. They performed an extensive study on the subcellular localization of LRP1 in U87 using confocal microscopy and electron microscopy combined

with immunolabeled U87 cells. They showed that LRP1 is present both in intracellular compartments and at the plasma membrane where LRP1 is largely confined to clathrin-coated pits. Supportive evidence on the role of LRP1 in the clathrin-coated pits has also been provided for other ligands of LRP1 such as CD44 where authors showed that the LRP-1-mediated uptake of CD44 occurs mainly through clathrin-coated pits¹⁰. Therefore, we have included additional supportive evidence in the discussion section.

8- The PWR data are opaque and not presented in a coherent molecular mechanistic manner. Given they are looking a small pieces of poorly characterized adherent membrane fragments, in the absence of any cytosolic support or source of cofactors, it is unclear what exactly it is they are reporting. The non-specialist reader should be considered when drafting these results and the accompanying discussion.

We agree with the reviewer that the way the data was presented was not very clear. We have made considerable efforts to substantially improve data clarity and ease the comprehension of the study for the reader as can be seen in this revised manuscript. Because of space constraints, we are not able to give full details about the PWR method. Nevertheless, we tried to be as précis and concise as possible. Moreover, we have included additional literature that further supports the study presented here.

Regarding the use of cell membrane fragments for the study of the ligand-induced conformational changes of the CXCR3 receptor, various studies with cell fragments by PWR have been reported, although the method use for the immobilisation in the PWR sensor was a little different from our study^{14,15}. The cell membrane fragments are derived from cells overexpressing CXCR3. They have been fully characterized by confocal microscopy for GFP and receptor content before immobilisation onto the silica surface. These fragments have a diameter of 3 μm (surface of 7.07 μm^2) to 4 μm (surface of 12.57 μm^2), are positive for GFP, and express CXCR3.

9- In Fig. 5E and G, it appears that different focal planes are being compared. In E, Si CTRL, looks like a medial optical section, while the SI LRP1 panels are ventral/basal sections to see PM/lamellapodia. This seems also the case in G.

We appreciate the reviewer's comments regarding confocal acquisition. A great attention on the focal plane was taken during confocal acquisition. To increase data visibility, we added a 3D projection of the cell transfected with SiRNA-CTRL or SiRNA-LRP1. This 3D projection is presented below (Figure 10 of the letter), and also in Figure S6B of the article.

Fig. 10: 3D projection of the cell transfected with SiRNA-CTRL or SiRNA-LRP1.

10- Are the authors arguing that beta-arrestins are not important for CXCR3 internalization and/or post-uptake signaling? There is an odd reference to some beta-arrestin data on page 15, but I do not see this referred to directly in the main body of the Results section. Any interplay between the beta-arrestins and LRP for clathrin-coated vesicle sorting of CXCR3 is left to the reader to interpret alone.

We have looked at the references page 15 related to beta-arrestins: the first reference of Meiser et al. show that in T lymphocytes CXCR3 can be internalized independently of clathrin and β -arrestin 1 and β -arrestin 2¹⁶, the two references that follow are related to the trans-golgi network^{17,18} and the last reference related to beta-arrestins is the reference of Lucker et al. that demonstrates that CXCR7 trafficking is regulated by beta-arrestin 2¹⁹. We did not include these references in the results part as it concerns the discussion part of the article. The odd reference the reviewer did mention could be the one of Uwada et al.²⁰ which was present in the reference section (reference 60) but which was not mentioned in the text. This is now corrected.

We agree with the reviewer that the interplay between beta-arrestins and LRP1 for clathrin-coated vesicle sorting of CXCR3 was not clearly mentioned in our study. However, we showed that beta-arrestins 1/2 are important for CXCR3 internalization as they are recruited at the cell membrane after 15 minutes of CXCR3 stimulation (Fig. S4A of the article). In fact, CXCR7 that has properties similar to those of CXCR3, is also regulated by beta-arrestin 2 and its internalization is clathrin-dependent¹⁹. They further demonstrated that CXCR7 was recycled to the cell membrane. To improve our discussion on beta-arrestins and to make a link with LRP1, we have included literature that further supports the study presented here. Goodman OB et al. showed that beta-arrestin functions as an adaptor in the receptor-mediated endocytosis pathway, and suggest a general mechanism for regulating the trafficking of G-protein-coupled receptors²¹. In addition, several studies on the role of LRP1 in the central nervous system have been performed and especially by analyzing the expression, subcellular distribution, and endocytic function of LRP1 in human glioblastoma U87 cells. Bu G. et al. showed that at the plasma membrane, LRP1 was largely confined to clathrin-coated pits and within cells, the protein partially colocalized within rough endoplasmic reticulum and the Golgi complex, similar results to ours¹¹. This has been now added to the manuscript. Additional support comes also from experimental results we obtained for this revision (see point 11).

11- Another issue is that the authors CANNOT claim that Pitstop 2 is a specific pharmacologic inhibitor of clathrin-mediated endocytosis. This compound has been thoroughly discredited and stops essentially all trafficking from the PM and also affects nuclear pore trafficking. This is published.

New experiments to analyse the clathrin-mediated endocytosis of CXCR3 have been carried out. Immunoblotting for clathrin, immunostaining for clathrin and CXCR3 and FACS analyses for CXCR3 were performed in U87-CXCR3-A cells knocked-down for the clathrin heavy chain. Our data showed that CXCR3 remains at the cell membrane when clathrin is depleted as evidenced by immunostaining and by FACS analysis for CXCR3. These data are presented in the Figure 11 of the letter and Figure 6-A,-B,-C, -D of the article.

Figure 11: Clathrin silencing in U87-CXCR3-A cells.

(A) Western blot for clathrin in U87-CXCR3-A cells treated with control siRNA (Si CTRL) or with a pool of 4 siRNA against human clathrin heavy chain (Si Clathrin). Tubulin used as a loading control. (B, C) Immunostaining in U87-CXCR3-A cells treated with Si CTRL or Si Clathrin (clathrin $\lambda=488\text{nm}$ and CXCR3 $\lambda=547\text{nm}$). All sections were observed under confocal laser-scanning microscope (Nikon eclipse Ti). Scale: 50 μm . Quantification performed on clathrin immunostaining. *** $p<0.001$. (D) FACS analyses and percentage of CXCR3 at the cell membrane in Si CTRL or Si Clathrin treated cells. ** $p<0.01$ and *** $p<0.001$.

Comments by reviewer 3:

MAJOR COMMENTS

1- In Figure 1A, authors described the CXCR3 mRNA copy number in several cell lines. They indicate, on one hand, that T98G expressed significantly more CXCR3A, and on the other hand, that U87 expressed significantly less CXCR3B. But compare to what?

Figure 1A of the article has been modified and PCR results are expressed as relative quantification. The U87 cell line has been used as the control cell line and CXCR3 mRNA expression of each cell line tested was compared to the U87 cell line.

2- In figure 1B, authors showed a western blot for total CXCR3 protein. The abundance of the protein did not fit with the relative gene expression showed in figure 1A. Is it due to the fact that western blot evaluate CXCR3 A+B?

As reliable CXCR3-isoforms specific antibodies are lacking, we did immunoblotting for total CXCR3 which recognize both isoforms.

3- In the same figure, authors are using vinculin as a loading control. As vinculin is a protein associated with focal adhesion plaques. Is it the relevant loading control, as CXCR3 seemed to be involved in invasion?

In figure 1B of the article, vinculin was used as a loading control. Indeed, this cytoskeletal protein is one of the major components for cell-cell and cell-matrix junctions. In the western blot, the whole cell lysate of Hek-293 and glioma cell lines cultured under unstimulated condition (resting conditions) that do not induce their invasion, was used. Under these conditions, vinculin expression levels are not modified. Of note, during invasion, vinculin may be modified with regard to its cellular localization²² and changes in expression levels during invasion have also been reported²³.

4- Authors showed nicely on CAM, that CXCR3A expressing U87 cells were more invasive than their control counterpart. Did they observe the same phenomenon with 1321N1 or NHATS cells expressing naturally more CXCR3?

We also considered broadening our study by using NHATS and 1321N1 that express endogenous CXCR3. However, in comparison to grade IV GBMs (GBM cell line such as U87), these cells are more difficult to culture *in vitro*. Furthermore, after implantation on the CAM, NHATS and 1321N1 cells were not able to form tumors in this experimental model. This may be also due to the time-constraints of the CAM model (< 21 days).

We, therefore, performed the CAM experiment with another GBM cell line, the T98G cell line. T98G cells express CXCR3 and respond to agonist stimulation (Fig. 1A, 1B and 8B of the article). These cells form smaller and more invasive tumors compared to the large angiogenic circumscribed U87 tumors (Figure 12 of the letter). Furthermore, migration of T98G is decreased when CXCR3 is depleted (see Figure 2 of the letter and Figure S2A of the article).

Figure 12: CAM assays were performed using U87 and T98G cells (n= 50 eggs for each condition). Seven days after implantation, images of CAM tumors were taken. As can be seen in this figure, U87 cells (left panels) form a well circumscribed angiogenic tumor, whereas T98G cells (right panels) exhibit only an infiltrative growth pattern.

5- Authors showed a close-colocalization of CXCR3 and LRP1 using STED HR microscope. They stated that the proteins were colocalized in the cytoplasm, near the nucleus (line 158). It's not obvious as no nuclear staining were used as a topological help. I strongly advise authors to include such a staining.

STED high resolution imaging was only performed on CXCR3 and LRP1 dual-color immunostaining. Nevertheless, in order to visualize nucleus, confocal staining with lower magnification was also performed on the same sample and superimposed. These results were added in the figure below (Figure 13 of the letter; Fig. S4B of the article).

Fig. 13: Visualization of nuclei in STED images

6- I advice authors to confirm their colocalization results (CXCR3-LRP1) by using Proximity Ligation Assay.

As the reviewer requested, we confirmed the colocalization between CXCR3 and LRP1 using the Proximity Ligation Assay technique. In order to validate our results, 3 negative control conditions have been done: 1/PLA in the HEK-CTRL cells that do not express CXCR3 and LRP1, 2/PLA in the HEK-CXCR3-A that express CXCR3-A but not LRP1 and 3/PLA using only the anti-LRP1 primary antibody in the U87-CXCR3-A cells. The PLA confirmed the specific colocalization of the two receptors in the U87-CXCR3-A cells (Figure 14 of the letter and Figure 3F of the article)

Figure 14: CXCR3/LRP1 colocalization using PLA. DAPI and CXCR3/LRP1 colocalization ($\lambda=546$ nm). *** $p<0.001$

7- I strongly suggest validating the localization of CXCR3 in the recycling compartment by immunogold techniques or correlative microscopy.

Immunogold localization of CXCR3 and syntaxin 6 has been performed in U87-CXCR3-A cells stimulated 60 min with the CXCR3 agonist PS372424. The data are presented in the Figure 4B of the article and figure 15 of the letter. Immunogold labeling of ultrathin cryosections of U87-CXCR3-A demonstrated that CXCR3 and syntaxin are both present in the trans-golgi (TG) network. These results confirm the localization of CXCR3 in the recycling compartment.

Figure 15 : Immunogold and Electron Microscopic Localization of CXCR3 and Syntaxin 6 in U87-CXCR3-A cells. TG: Trans Golgi

8- One-way ANOVA followed with multiple comparisons were adequately used. However, I strongly suggest the use of Bonferroni post-test instead of Tukey.

As requested by the reviewer, analyses of multiple comparisons were modified and performed with one-way analysis of variance, followed by Bonferroni post hoc tests.

MINOR COMMENTS

1- In Figure 1A, authors described the CXCR3 mRNA copy number in several cell lines. I am however disturbed by the use of arbitrary unit. I guess it is a ratio between CXCR3 gene expression and the reference gene expression. In that case, CXCR3 copy number should be replaced by a relative gene expression.

We have corrected and replaced by relative gene expression.

2- Boyé et al stably transfected U87 cells with GFP-CXCR3-A encoding vectors. They showed by western blot the expression of this fusion-protein using GFP antibody. Results will be more convincing by using anti-CXCR3 antibody.

U87-CXCR3-A cells express the full length of CXCR3-A isoform and western blots using anti-CXCR3 antibody have been performed in these cells (Fig. S1E of the article). The western blot using GFP antibody was done in cell lysate of Hek-293 cells that have been stably transfected with the full length CXCR3-A tagged with GFP in the C-terminal domain. Therefore, blotting with the anti-GFP antibodies was done to specifically detect CXCR3-A because the available anti-CXCR3 antibodies do not distinguish between CXCR3-A and CXCR3-B. In HEK-CXCR3-A cells, FACS analyses were performed using CXCR3 antibodies coupled to PE (Fig. S1F of the article). This demonstrated high CXCR3 expression levels in these cells.

3- In the Figure S1A, authors indicated that CXCR3B is not expressed in U87 after CXCR3-A transfection. They should replace “NS” (not significant) by “ND” (not detected).

We have corrected in the Figure S1B (former figure S1A) and replaced NS by ND.

4- In the Figure S1B, they showed the CXCR3 protein abundance after siRNA transfection. CXCR3-9 siRNA worked pretty well (90% depletion) but it was not the case of the 3 others siRNA. Do the authors have a comment on this result? Statistically, I'm surprised that CXCR3-9 siRNA depletion was less significant than the other depletion.

We have corrected in Figure S1B. Indeed there are 2 stars ** for SiRNA CXCR3-11 et -12 and 3 stars *** for SiRNA CXCR3-9.

5- In figure S1C legend, they should replace “pv21” by “pV21”.

We have corrected in the legend of figure S1D (former figure S1C) and replaced pv21 by pV21.

6- In figure S1D legend, they should replace “Western” by “western” as western, at the opposite of Southern, is not a family name. Same comment for lines 106 and 241.

We have corrected in the Figure S1D in the legend and in the text and replaced Western by western.

7- Authors used the CAM model to generate U87 tumors expressing CXCR3. Ideally they should prove that CXCR3 antibody does not cross react with a chicken protein. Did they try to grow HEK tumors on CAM and observe the same result?

The antibody anti-CXCR3 (CXCR3 mouse monoclonal 2 ar1) has been tested on the CAM model alone and showed no reactivity for chicken tissue (Figure 16 of the letter). Regarding the Hek-293 cells, this cell line is embryonic and is not a tumor cell line, therefore these cells have not the property to form a tumor *in vivo* in the CAM model.

Figure 16: Immunostaining for CXCR3 (antibody 2ar1) in the chicken CAM alone and in the CAM implanted with U87-CXCR3-A cells. Merge images with dapi and CXCR3 ($\lambda= 488$ nm).

8- Figure2C should have the same x-axis labeling.

This figure has been corrected with regard to the x-axis labelling.

9- Authors stated that CXCR3 ligands (PS372424, CXCL10 & CXCL4L1) increased Erk phosphorylation in their CXCR3-expressing U87. Their results are obvious but will be more convincing by adding quantification.

The requested quantification has been added in the Figure S2F of the article. In addition the quantification for the Figure 2H of the article has also been added in the Figure S2G of the article.

10- CXCR3 and LRP1 were partially colocalized by STED HR microscopy using metamorph software (line 160). Authors should indicate what was the procedure used: Pearson, Mander, Li, ...

In order to quantify the colocalization of CXCR3 and LRP1 per cell with STED high - resolution microscopy, we based our analysis on the detection of colocalized pixels for CXCR3 and LRP1. We calculated the global intensity for each fluorophore per cell. For each image, we created regions of interest (ROI) and calculated the global intensity for each fluorophore. Colocalization was estimated by overlapping of the ROIs of the two fluorophores (CXCR3 and LRP1). The percentage of overlapping between the two fluorophore was normalized to the global intensity of each fluorophore (expressed in percentage). The analysis was done in the Bordeaux Imaging Center (BIC) using MetaMorph software.

11- Authors used RAP in order to antagonize LRP1 ligand binding. They observed by coIP a decrease of the CXCR3:LRP1 complex formation. I notice a decrease of LRP1beta abundance in the complex. This decrease seemed proportional to the CXCR3 decrease. Do the authors have considered a specific interaction of CXCR3 and LRP1beta?

We have used mini-receptors of LRP1 that only contains the ligand binding domain IV (HA-LRPIV, Fig. 4B of the article) and other constructs that either contain one of the other domains (domain I, II and III) or the beta chain alone. We could only co-immunoprecipitate HA-LRPIV in a reproducible fashion, but not any of the other domains.

12- Authors should replace “Down-regulation of LRP-1 expression” (line 225) by “LRP-1 depletion”.

We have corrected and replaced down-regulation of LRP-1 expression by LRP-1-depletion in the text.

13- In figure 5A, authors showed the result of the LRP1 siRNA transfection in U87 cells. The U87-CTRL western-blot does not fit (see siRNA LRP1-1) with the quantification presented in the graphic.

The quantification in figure 7A right panel (former figure 5A) corresponds to the 3 independent experiments. The western blot on the left panel, represents a single experiment.

14- Line 230, authors wrote: higher shifts in p- than s- polarisation are observed. IN the corresponding figure (5B), they should use the same y-axis range in order to allow the comparison. I did not understand the link between the Figure 5B and the table below.

The figure 5B (now Figure 7B of the article) has been now been changed so that the same y-axis range is used so that data comparison is easier.

Indeed, the figure and legend was not very clear making it difficult to establish a link between the panels and the table below, we have now revised to make it more clear. The table below the panels shows the magnitude of the response, the maximum PWR spectral shift (in millidegrees) observed at saturation obtained with p- and s-polarized light for both the data shown above of the siLRP1 treated cells and the siCTRL that is not presented. The table has been modified as well as the legend to make it clearer. This data is also now better discussed in the article (page 11).

15- In all their figures, authors used personalized symbols to indicate statistical significance. It is more pertinent to use: one star for $p < 0.05$, two stars for $p < 0.01$ and 3 stars for $p < 0.001$.

We have corrected and replaced in all figures one star for $p < 0.05$, two stars for $p < 0.01$ and 3 stars for $p < 0.001$.

16- In figure 6E the scale bar represents $10\mu\text{m}$ when the legend indicates $5\mu\text{m}$.

We have corrected in the legend of the figure 6E (now Figure 8E of the article), the scale was indeed $10\mu\text{m}$.

References

- 1 Maru, S. V. *et al.* Chemokine production and chemokine receptor expression by human glioma cells: role of CXCL10 in tumour cell proliferation. *Journal of neuroimmunology* **199**, 35-45, doi:10.1016/j.jneuroim.2008.04.029 (2008).
- 2 Pu, Y. *et al.* High expression of CXCR3 is an independent prognostic factor in glioblastoma patients that promotes an invasive phenotype. *Journal of neuro-oncology* **122**, 43-51, doi:10.1007/s11060-014-1692-y (2015).
- 3 Sharma, I., Siraj, F., Sharma, K. C. & Singh, A. Immunohistochemical expression of chemokine receptor CXCR3 and its ligand CXCL10 in low-grade astrocytomas and glioblastoma multiforme: A tissue microarray-based comparison. *Journal of cancer research and therapeutics* **12**, 793-797, doi:10.4103/0973-1482.153657 (2016).
- 4 Ponten, J. & Macintyre, E. H. Long term culture of normal and neoplastic human glia. *Acta pathologica et microbiologica Scandinavica* **74**, 465-486 (1968).
- 5 Sasai, K., Akagi, T., Aoyanagi, E., Tabu, K., Kaneko, S. and Tanaka, S. O6-methylguanine-DNA methyltransferase is downregulated in transformed astrocyte cells: implications for anti glioma therapies. *Mol Cancer*. **6**, 36 (2007).
- 6 Allen, M., Bjerke, M., Edlund, H., Nelander, S. & Westermarck, B. Origin of the U87MG glioma cell line: Good news and bad news. *Science translational medicine* **8**, 354re353, doi:10.1126/scitranslmed.aaf6853 (2016).
- 7 Stein GH. T98G: an anchorage-independent human tumor cell line that exhibits stationary phase G1 arrest in vitro. *J Cell Physiol*. **99**:43-54 (1979).
- 8 Jenh, C. H. *et al.* A selective and potent CXCR3 antagonist SCH 546738 attenuates the development of autoimmune diseases and delays graft rejection. *BMC immunology* **13**, 2, doi:10.1186/1471-2172-13-2 (2012).

- 9 Quemener, C. *et al.* Dual roles for CXCL4 chemokines and CXCR3 in angiogenesis and invasion of pancreatic cancer. *Cancer research*, doi:10.1158/0008-5472.CAN-15-2864 (2016).
- 10 Perrot, G. *et al.* LRP-1--CD44, a new cell surface complex regulating tumor cell adhesion. *Molecular and cellular biology* **32**, 3293-3307, doi:10.1128/MCB.00228-12 (2012).
- 11 Bu, G., Maksymovitch, E. A., Geuze, H. & Schwartz, A. L. Subcellular localization and endocytic function of low density lipoprotein receptor-related protein in human glioblastoma cells. *The Journal of biological chemistry* **269**, 29874-29882 (1994).
- 12 Wolf, B. B., Lopes, M. B., VandenBerg, S. R. & Gonias, S. L. Characterization and immunohistochemical localization of alpha 2-macroglobulin receptor (low-density lipoprotein receptor-related protein) in human brain. *The American journal of pathology* **141**, 37-42 (1992).
- 13 Fack, F. *et al.* Bevacizumab treatment induces metabolic adaptation toward anaerobic metabolism in glioblastomas. *Acta neuropathologica* **129**, 115-131, doi:10.1007/s00401-014-1352-5 (2015).
- 14 Georgieva, T. *et al.* Unique agonist-bound cannabinoid CB1 receptor conformations indicate agonist specificity in signaling. *European journal of pharmacology* **581**, 19-29, doi:10.1016/j.ejphar.2007.11.053 (2008).
- 15 Salamon, Z. *et al.* Plasmon-waveguide resonance studies of ligand binding to integral proteins in membrane fragments derived from bacterial and mammalian cells. *Analytical biochemistry* **387**, 95-101, doi:10.1016/j.ab.2009.01.019 (2009).
- 16 Meiser, A. *et al.* The chemokine receptor CXCR3 is degraded following internalization and is replenished at the cell surface by de novo synthesis of receptor. *Journal of immunology* **180**, 6713-6724 (2008).
- 17 Koliwer, J., Park, M., Bauch, C., von Zastrow, M. & Kreienkamp, H. J. The golgi-associated PDZ domain protein PIST/GOPC stabilizes the beta1-adrenergic receptor in intracellular compartments after internalization. *The Journal of biological chemistry* **290**, 6120-6129, doi:10.1074/jbc.M114.605725 (2015).
- 18 Shewan, A. M. *et al.* GLUT4 recycles via a trans-Golgi network (TGN) subdomain enriched in Syntaxins 6 and 16 but not TGN38: involvement of an acidic targeting motif. *Molecular biology of the cell* **14**, 973-986, doi:10.1091/mbc.E02-06-0315 (2003).
- 19 Luker, K. E., Steele, J. M., Mihalko, L. A., Ray, P. & Luker, G. D. Constitutive and chemokine-dependent internalization and recycling of CXCR7 in breast cancer cells to degrade chemokine ligands. *Oncogene* **29**, 4599-4610, doi:10.1038/onc.2010.212 (2010).
- 20 Uwada, J., Yoshiki, H., Masuoka, T., Nishio, M. & Muramatsu, I. Intracellular localization of the M1 muscarinic acetylcholine receptor through clathrin-dependent constitutive internalization is mediated by a C-terminal tryptophan-based motif. *Journal of cell science* **127**, 3131-3140, doi:10.1242/jcs.148478 (2014).
- 21 Goodman, O. B., Jr. *et al.* Beta-arrestin acts as a clathrin adaptor in endocytosis of the beta2-adrenergic receptor. *Nature* **383**, 447-450, doi:10.1038/383447a0 (1996).
- 22 Giannone, G. Super-resolution links vinculin localization to function in focal adhesions. *Nature cell biology* **17**, 845-847, doi:10.1038/ncb3196 (2015).
- 23 Coll, J. L. *et al.* Targeted disruption of vinculin genes in F9 and embryonic stem cells changes cell morphology, adhesion, and locomotion. *Proceedings of the National Academy of Sciences of the United States of America* **92**, 9161-9165 (1995).

REVIEWERS' COMMENTS:

Reviewer #1 (Remarks to the Author):

Revised manuscript adequately addresses critiques raised in earlier review.

Reviewer #4 (Remarks to the Author):

The authors have responded thoroughly to Reviewer 2's concerns, with a number of important new experiments. There are 3 minor points that remain – these do not require new data, but attention to the wording.

Comment 4: The antibodies used have been now identified in some figures, but not the key figure -- staining of glioblastoma samples. Also, the wording remains too strong – the data in Table 2 suggest that there is an approximately 50% reduction in LRP1 in the invasive regions compared to the angiogenic regions, which is not consistent with LRP1 being “totally shut down” or “extinction of LRP1.”

Comment 6: The authors seem to misunderstand the reviewer. In the manuscript, it seems that the authors are claiming that ligand stimulation causes recruitment of CXCR3 and LRP1 to the plasma membrane. If they are not making this claim, they need to reword the statement on p. 7 line 160 (and potentially chose a more representative image in Fig S4C). If they are making this claim, they need to rationalize it.

Comment 11: The addition of clathrin knockdown is very important. Still, the authors should mention the caveats to Pitstop2 in the manuscript.

Comments by Reviewer 1:

Revised manuscript adequately addresses critiques raised in earlier review.

Comments by Reviewer 2:

The authors have responded thoroughly to Reviewer 2's concerns, with a number of important new experiments. There are 3 minor points that remain – these do not require new data, but attention to the wording.

Comment 4: The antibodies used have been now identified in some figures, but not the key figure -- staining of glioblastoma samples. Also, the wording remains too strong – the data in Table 2 suggest that there is an approximately 50% reduction in LRP1 in the invasive regions compared to the angiogenic regions, which is not consistent with LRP1 being “totally shut down” or “extinction of LRP1.”

Reply: As the reviewer requested, we have now indicated the antibodies used in the legend of Fig. 9. Furthermore, we have changed the wording and replaced shut-down and extinction by is strongly reduced and inhibition of LRP1 expression (page 17, lines 396 and 399 in Word Document; page 18, lines 405 and 408 in merged pdf).

Comment 6: The authors seem to misunderstand the reviewer. In the manuscript, it seems that the authors are claiming that ligand stimulation causes recruitment of CXCR3 and LRP1 to the plasma membrane. If they are not making this claim, they need to reword the statement on p. 7 line 160 (and potentially chose a more representative image in Fig S4C). If they are making this claim, they need to rationalize it.

Reply: In the manuscript, we do not make the claim of that ligand stimulation induces recruitment, but that upon ligand stimulation the Pearson coefficient increases and that more CXCR3:LRP1 complexes are formed. It is clear that upon ligand stimulation more CXCR3 is found at the plasma membrane. This may be due to relocalization of CXCR3 to plasma membrane. As the fluorescent tag is fused in frame to the carboxyl terminus of CXCR3-A in the HEK-CXCR3-A cells, receptor trafficking was followed dynamically by time-lapse microscopy upon stimulation with CXCR3 agonist (PS372424), CXCL10 and CXCL4L1. Before ligand stimulation, CXCR3-A isoform was localized mainly in the cytoplasm near to the nucleus (Fig 1 of this reply). Upon ligand stimulation, CXCR3-A was more homogenously distributed within the cell and increased on the cell membrane. The kymograph, below each condition, gives a graphical representation of the spatial position of CXCR3-A overtime. These images clearly show that upon stimulation with various ligands, the receptor is partially relocalized, even if transitorily when internalization follows. The molecular mechanisms of this relocalization are not known at the present time. This is stated now page 15 lines 352-353 in Word document; page 16 lines 362- 363 in merged pdf).

Figure 1: Cytoplasm and membrane relocation of CXCR3-A ($\lambda=488$ nm) upon CXCR3 agonist stimulation (PS372424, CXCL10, CXCL4L1) were shown by time-lapse microscopy (5 min) in HEK-CXCR3-A cells (HEK-3A). NS: unstimulated condition, DMSO: negative control. Sections were observed at 630X magnification. Kymograph analyses were performed using ImageJ software.

Comment 11: The addition of clathrin knockdown is very important. Still, the authors should mention the caveats to Pitstop2 in the manuscript.

Reply: This has been added in the text of the manuscript (page 10 lines 214-216 in word document; page 10, lines 221-223 in merged pdf).